# Geometry-Aware Dataset Condensation for Diffusion Model Training

Xiao Cui [1]  Yulei Qin [2]  Mo Zhu [3]  Wengang Zhou [1]  Hongsheng Li [4]  Houqiang Li [1]

## Abstract

Dataset condensation aims to construct compact datasets from real data via synthesis or selection. However, existing approaches are ill-suited for diffusion model training: synthetic data generation often yields low-fidelity samples unsuitable for authentic modeling, while real subset selection typically fails to preserve the distributional geometry required by diffusion likelihood objectives. To address this, we propose to reformulate real subset selection as a geometry-aware distribution alignment problem. By incorporating one-sided partial optimal transport, our method selectively aligns a compact subset with the full data distribution while allowing unmatched mass in low-density regions, ensuring the preserved geometric structure necessary for effective diffusion model training. To further ensure distributional fidelity, we complement geometric alignment with lightweight feature-statistics and semantic consistency regularization. An efficient two-stage discrete optimization strategy is proposed to achieve this alignment objective. Extensive experiments across diffusion variants, subset sizes, image resolutions, and training rounds show that our method achieves superior fidelity and distributional coverage in diffusion model training.

## 1. Introduction

Deep learning has achieved remarkable success across discriminative and generative paradigms, largely driven by massive annotated datasets (Yan et al., 2025; Cui et al., 2021; Qin et al., 2025). However, reliance on large-scale data incurs substantial storage and computational costs, especially in resource-constrained settings (Liu & Du, 2025; Yu et al., 2023). Dataset condensation addresses this chal-

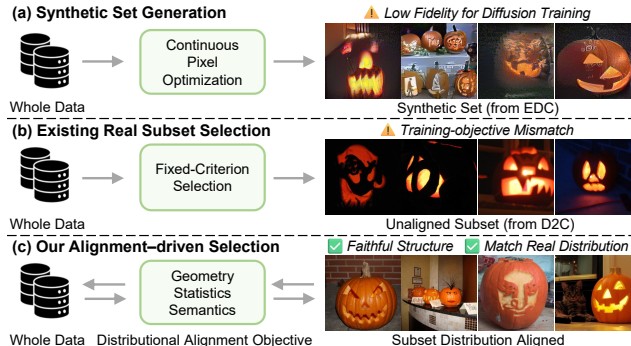

*Figure 1.* Comparison of dataset condensation methods. (a) Synthetic set generation methods create synthetic samples through continuous pixel optimization, often resulting in low-fidelity samples that are unsuitable for diffusion training. (b) Existing real subset selection methods use fixed criteria to select real samples but do not align the subset with the full data distribution. (c) Our alignment-driven selection optimizes for distributional alignment, preserving structure and matching the real data distribution.

lenge by constructing more compact datasets through *data synthesis* or *data selection*, while preserving the essential knowledge of the full dataset (Geng et al., 2023; Sachdeva & McAuley, 2023). For training diffusion models, such condensation approaches are particularly important as the modeling of high-quality samples by diffusion models often requires extremely expensive or even prohibitive training in terms of both data and compute. It is therefore practically important to enable diffusion generative modeling on condensed datasets, which remains underexplored.

Most existing condensation methods are developed for recognition tasks such as classification (Shao et al., 2024; Wang et al., 2025b; Liu et al., 2023a; Cui et al., 2025a), and their optimization objectives are not aligned with diffusion training, which fundamentally targets modeling the data distribution rather than decision boundaries. In addition, a large portion of these methods instantiate condensation via *data synthesis*, where pixel-level synthesis is performed with gradient-based optimization. Such continuous synthesis, while beneficial for discriminative tasks, is prone to high variance and introduces noise that distorts fine-grained structures and distributional characteristics. As a result, synthetic methods are fundamentally ill-suited for diffusion model training, as diffusion models are highly sensitive to noise and structural distortions under extreme data budgets.

[1]University of Science and Technology of China [2]Independent Researcher [3]Zhejiang University [4]CUHK MMLAB. Correspondence to: Houqiang Li <lihq@ustc.edu.cn>, Wengang Zhou <zhwg@ustc.edu.cn>.

*Proceedings of the 43rd International Conference on Machine Learning*, Seoul, South Korea. PMLR 306, 2026. Copyright 2026 by the author(s).

Alternatively, a small number of works (Zheng et al., 2023; Zhou et al., 2023; Zhao et al., 2024) consider *data selection* for dataset condensation. From a data perspective, such approaches are inherently more promising for diffusion model training, as they retain the high-fidelity structure of real samples. However, as illustrated in Fig. 1, these methods select real subsets according to fixed or heuristic criteria, rather than optimizing a task-aligned objective. In particular, the only prior work that explicitly targets diffusion model training, D$^2$C (Huang et al., 2026), ranks images within each class by their diffusion difficulty and selects samples at equal intervals along this one-dimensional difficulty axis to construct the condensed subset. While effective to some extent, such a fixed-criterion strategy suffers from two fundamental limitations. First, many existing selection strategies are not derived from an explicit, training-aligned optimization objective, but instead rely on heuristic or proxy criteria for subset selection. Even when an objective is specified, it is often only loosely related to diffusion training, which is fundamentally driven by likelihood maximization and therefore calls for distributional rather than scalar alignment. Second, even under the assumption of an ideal, training-aligned objective, the one-pass greedy selection strategies adopted by most existing methods generally lack the capacity to reliably optimize it in the discrete combinatorial space of subset selection. As a result, they are unable to perform principled distributional alignment, leading to subsets that can be poorly aligned with the full data distribution.

To overcome these limitations, we cast dataset condensation for diffusion training as a geometry-aware distribution alignment problem subject to a discrete selection constraint. Here, "geometry" refers to the distributional support geometry in representation space. In the context of diffusion-based generative modeling, we aim to answer the following research questions: *(i) How can we achieve principled distribution matching in the feature space?* We utilize optimal transport (OT) to offer a principled mechanism for matching the selected subset to the full data distribution in feature space, capturing both global coverage and local geometric structure. *(ii) How can we mitigate alignment degradation caused by severe capacity mismatches?* The subset size budget is drastically smaller than that of the full dataset, leading to a severe capacity mismatch. Under this regime, enforcing balanced matching of the entire set is uninformative during OT as transport mass is inevitably spread across peripheral, low-density regions. We propose the one-sided partial OT (POT) with target-side capacity constraints to allow unmatched mass and sharpen alignment toward high-density, geometrically stable, dominant regions. *(iii) To what extent do auxiliary constraints enhance OT-based geometry shaping?* We propose lightweight statistical and semantic regularization to complement OT-based geometry matching by promoting global distributional fidelity and

semantic consistency of the selected subset. Together, these three components form a unified geometry-aware distribution alignment objective.

Optimizing this alignment objective over subset selection is inherently combinatorial, since the decision variable is a fixed-size subset rather than continuous parameters. To make the problem tractable at scale, we introduce a two-stage discrete optimization strategy that mirrors the needs of diffusion-oriented alignment: a geometry-guided greedy construction that quickly establishes broad manifold coverage under the POT-driven objective, followed by a swap-based refinement that corrects early myopic choices through targeted exchanges between selected and unselected samples. This solver directly optimizes the same alignment objective defined above, yielding compact subsets that remain faithful to both the geometric structure and the distributional profile required for effective diffusion model training. For efficiency, we compute the POT cost via entropic Sinkhorn iterations in a batch form, enabling scalable GPU computing through parallel scheduling and memory balancing.

In summary, our contributions are: (1) Based on the diffusion training objective, we reformulate dataset condensation for diffusion models as a distribution alignment problem, where the condensed subset is selected by optimizing a one-sided POT objective with statistical regularization. (2) We propose a two-stage discrete optimization framework that efficiently solves this alignment problem, moving beyond fixed-criterion or ranking-based sampling. (3) Extensive experiments show that our method consistently outperforms prior approaches across various diffusion variants, subset sizes, training steps, and evaluation protocols, achieving superior fidelity and efficiency in diffusion training.

## 2. Related Work

**Dataset Condensation.** Existing dataset condensation methods fall into two optimization paradigms (Wu et al., 2025a): synthetic set generation and real subset selection.

*Synthetic Set Generation.* These methods directly optimize a compact synthetic dataset in a continuous space to approximate the learning behavior of the full dataset. Distribution-matching (Zhao & Bilen, 2023; Wang et al., 2025b; Li et al., 2025b; Cui et al., 2025b), trajectory-matching (Cazenavette et al., 2022; Zhong et al., 2025; Wang et al., 2025a; Zhao et al., 2025), gradient-matching (Zhao et al., 2020; Kim et al., 2022; Liu et al., 2023a;b), and model-inversion-based methods (Yin et al., 2023; Cui et al., 2025a; Shen et al., 2025; Hu et al., 2025; Cui et al., 2026a) all rely on differentiable optimization of pixel values to minimize surrogate objectives. However, such optimization, unconstrained by the real data manifold, often yields low-fidelity and mode-collapsed samples, making these methods unsuitable for generative

training tasks that require high-fidelity data. Generative-model-based methods (Chen et al., 2025; Moser et al., 2025; Wu et al., 2025b; Cui et al., 2026b) rely on pretrained generators to produce images. Training these generators is computationally expensive, and their learned priors are often difficult to adapt across domains. Moreover, the generated samples inevitably deviate from real image distributions, making them less effective than directly selecting real images for diffusion model training.

*Real Subset Selection.* Discrete condensation methods select informative real samples instead of synthesizing new ones. The K-center method (Gonzalez, 1985; Sener & Savarese, 2018), with its farthest-point greedy criterion, tends to push selected samples toward the embedding-space boundary, overemphasizing extreme and low-density regions, while herding (Welling, 2009) overfits to the global mean and lacks sufficient coverage of mid- and long-tail variations. CCS (Zheng et al., 2023) further strengthens coverage, yet it still operates as a one-shot, criterion-driven selection scheme. Dataset quantization methods (Zhou et al., 2023; Zhao et al., 2024; Li et al., 2025a) discretize data into bins via submodular surrogates, but this binning collapses intra-bin geometry into coarse prototypes and introduces irreversible quantization bias. Beyond pure selection, methods such as RDED (Sun et al., 2024a) and OD3 (Khatib et al., 2025) further adapt distilled datasets to specific tasks through sample cropping and instance placement. $D^2C$ (Huang et al., 2026) is the first work to adapt dataset condensation for diffusion model training by proposing difficulty-guided selection. However, this one-dimensional ranking collapses the inherently multi-mode data distribution and ignores the manifold structure underlying the data distribution. In contrast, our method formulates real subset selection as a geometry-aware distribution alignment problem in the representation space. By incorporating one-sided POT and statistical regularization within a two-stage optimization framework, our approach preserves both geometric and distributional structures of real data while remaining fully discrete and computationally efficient.

**Efficient Diffusion Model Training.** Recent studies have explored various strategies to improve the efficiency of diffusion model training from the model side. Optimization-based approaches refine the noise schedule (Ho et al., 2020; Song et al., 2020; Karras et al., 2022), employ adaptive loss weighting (Nichol & Dhariwal, 2021; Hang et al., 2023), introduce OT-based training objectives (Kim et al., 2024), or distill the denoising trajectory into fewer steps (Salimans & Ho, 2022; Song et al., 2023) to accelerate convergence. Architectural improvements, including lightweight U-Net variants (Rombach et al., 2022; Saharia et al., 2022) and transformer-based diffusion backbones (Peebles & Xie, 2023; Ma et al., 2024), further enhance scalability and parallelism. More recently, representation alignment methods

such as REPA (Yu et al., 2025) and REPA-E (Leng et al., 2025) introduce auxiliary supervision that aligns intermediate diffusion features with pretrained visual encoders, improving convergence speed and stability. Although these approaches effectively reduce computational cost, they all operate on the model side and still assume access to large, diverse datasets. In contrast, we focus on dataset condensation, a complementary data-centric approach that optimizes the training data to improve efficiency. By constructing compact yet geometry-aligned subsets of real images, we reduce storage cost and accelerate diffusion model training, providing an orthogonal path to model-side methods.

## 3. Methods

### 3.1. Problem Statement

We study the problem of dataset condensation for diffusion model training via real subset selection. Given a large dataset $\mathcal{T} = \bigcup_{c=1}^{C} \mathcal{T}_c$, where $\mathcal{T}_c$ denotes class $c$ samples, the goal is to **select** a compact, **real** subset $\mathcal{S} = \bigcup_{c=1}^{C} \mathcal{S}_c$, with $\mathcal{S}_c \subset \mathcal{T}_c$ and $|\mathcal{S}_c| \ll |\mathcal{T}_c|$, using equal subset size across classes. A diffusion model trained on $\mathcal{S}$ should achieve competitive generative performance with limited training iterations.

### 3.2. From Ranking-based to Geometry-aware Selection

Diffusion models are trained by maximizing data likelihood through a variational objective, where optimizing the ELBO enforces distribution-level matching between the learned generative process and the underlying data distribution. Unlike discriminative objectives that emphasize decision boundaries, diffusion training is inherently sensitive to distortions of the data support itself, as likelihood estimation depends on preserving relative relationships and local structure rather than reducing samples to scalar scores.

When training data are condensed, the selected subset must therefore remain well aligned with the original data distribution, as misalignment directly translates into a biased likelihood objective. This requirement exposes a fundamental limitation of ranking-based selection strategies. The only existing method tailored for diffusion training, $D^2C$ (Huang et al., 2026), projects samples onto a scalar difficulty axis derived from diffusion loss and performs selection by uniformly spacing along this one-dimensional ranking. While such a score provides a coarse proxy for training difficulty, it collapses rich high-dimensional distributional information into a single scalar and does not explicitly optimize distribution alignment. As a result, samples from distinct modes may receive similar scores, while structurally related samples can be separated along the ranking, leading to subsets that are suboptimal for diffusion training.

These observations suggest that diffusion-oriented dataset

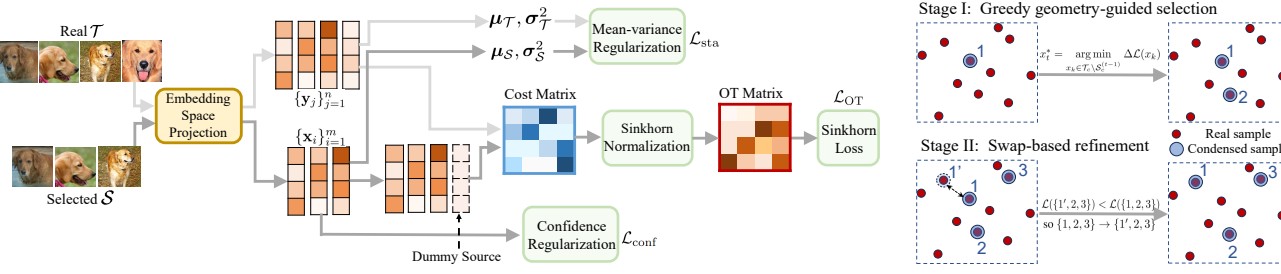

*(a)* Distribution alignment objective defined by one-sided POT and statistical regularization.    *(b)* Two-stage optimization strategy.

*Figure 2.* Illustration of the proposed pipeline. Mean–variance regularization aligns real and selected embeddings, and confidence regularization ensures semantic reliability. A dummy-source-augmented cost matrix is normalized via Sinkhorn iterations to obtain the one-sided POT alignment. The total objective, composed of these three components, is optimized using a two-stage strategy.

condensation should be formulated as an explicit distribution alignment problem in the representation space. Specifically, two requirements naturally arise from the ELBO objective: (1) *geometric alignment*, as ELBO-based diffusion training is sensitive to structural bias in the learned data support, and (2) *distributional fidelity*, since likelihood maximization requires preserving the global diversity and semantic balance of the data distribution. To jointly satisfy these requirements, we formulate dataset condensation as a geometry-aware transport alignment problem between the selected subset and the full dataset in the learned feature space. By leveraging one-sided POT with statistical regularization, our approach provides a principled and task-aligned mechanism for selecting representative real subsets. Notably, our framework is agnostic to both the feature encoder and classifier, as detailed in Appendix G.3.

We propose an alignment-driven discrete optimization framework. In the distribution alignment module (Fig. 2a), one-sided POT and statistical regularization define the objective, which is optimized via a two-stage discrete strategy (Fig. 2b); the following subsections detail each component.

### 3.3. One-sided Partial Optimal Transport

Following previous works (Yin et al., 2023; Huang et al., 2026; Zhou et al., 2023), we perform condensation independently for each class to ensure balanced coverage and efficiency. Let $\mathcal{S}_{c(e)} = \{\mathbf{x}_i\}_{i=1}^m$ and $\mathcal{T}_{c(e)} = \{\mathbf{y}_j\}_{j=1}^n$ denote the feature embeddings of the selected subset and the full real set for the class c. The pairwise transport cost is defined as the squared Euclidean distance:

$$\mathbf{C}_{ij} = \|\mathbf{x}_i - \mathbf{y}_j\|_2^2. \tag{1}$$

Classical balanced OT enforces full mass alignment by:

$$\min_{\boldsymbol{\pi} \geq 0} \langle \mathbf{C}, \boldsymbol{\pi} \rangle \quad \text{s.t.} \quad \boldsymbol{\pi} \mathbf{1}_n = \boldsymbol{\mu}, \ \boldsymbol{\pi}^\top \mathbf{1}_m = \boldsymbol{\nu}, \tag{2}$$

where $\boldsymbol{\pi} \in \mathbb{R}_+^{m \times n}$ denotes the *transport plan* that specifies the mass transported from each source sample $\mathbf{x}_i$ to each

target sample $\mathbf{y}_j$. The cost matrix $\mathbf{C}$ encodes the pairwise distances between source and target embeddings, and $\boldsymbol{\mu}$ and $\boldsymbol{\nu}$ are the source and target marginals, typically uniform distributions. However, this rigid constraint is misaligned with diffusion-oriented subset selection, as it enforces symmetric full mass matching between real and selected samples. In diffusion training, the selected set serves as a training distribution optimized for likelihood approximation rather than a reconstruction target, making full alignment unnecessary and biasing the selected set away from the core manifold. Alignment should therefore be relaxed, allowing transport mass to concentrate on the core manifold.

To enable selective matching, we employ a one-sided partial optimal transport formulation, where the source mass is still fully transported while the target side is relaxed to be satisfied under a capacity constraint:

$$\min_{\boldsymbol{\pi} \geq 0} \langle \mathbf{C}, \boldsymbol{\pi} \rangle \quad \text{s.t.} \quad \boldsymbol{\pi} \mathbf{1}_n = \boldsymbol{\mu}, \quad \boldsymbol{\pi}^\top \mathbf{1}_m \leq \kappa \bar{\boldsymbol{\nu}}, \tag{3}$$

where $\boldsymbol{\mu} = \frac{1}{m}\mathbf{1}$, $\bar{\boldsymbol{\nu}} = \frac{1}{n}\mathbf{1}$ is the reference marginal on the target side, and $\kappa$ is a scalar capacity scaling factor. When $\kappa = 1$, Eq. (3) reduces to balanced OT; for $\kappa > 1$, the inequality allows part of the target mass to remain unmatched, concentrating transport on informative regions.

**Dummy-source reformulation.** To solve Eq. (3) efficiently, we reformulate it into a balanced optimal transport problem via the *dummy-source trick* (Chapel et al., 2020), as detailed in Appendix B. We first augment the cost matrix as

$$\mathbf{C}_{\mathrm{aug}} = \begin{bmatrix} \mathbf{C} \ ; \ \delta \mathbf{1}_n^\top \end{bmatrix}, \qquad \delta > 0. \tag{4}$$

where $\delta$ denotes a dummy-source cost introduced to ensure stable and differentiable computation, $[\cdot \, ; \, \cdot]$ denotes vertical concatenation. In practice we set $\delta = \mathrm{median}(\mathbf{C}) \cdot \gamma$. We then introduce an augmented transport plan $\boldsymbol{\Pi} \in \mathbb{R}_+^{(m+1) \times n}$ with an additional dummy row supplying $s = \kappa - 1$ mass, which absorbs unused target capacity. We further apply entropic regularization to provide a smooth and convex ap-

proximation to the OT objective:

$$\min_{\mathbf{\Pi} \geq 0} \langle \mathbf{C}_{\text{aug}}, \mathbf{\Pi} \rangle - \varepsilon H(\mathbf{\Pi}) \quad \text{s.t. } \mathbf{\Pi}\mathbf{1} = [\boldsymbol{\mu}; s], \ \mathbf{\Pi}^{\top}\mathbf{1} = \kappa\bar{\boldsymbol{\nu}}. \tag{5}$$

Here, the entropic regularization term $H(\mathbf{\Pi})$, weighted by $\varepsilon$, enables efficient optimization via Sinkhorn iterations.

**Sinkhorn solver.** We define the Gibbs kernel $\mathbf{K} = \exp(-\mathbf{C}_{\text{aug}}/\varepsilon) \in \mathbb{R}^{(m+1)\times n}$, which encodes pairwise affinities derived from the augmented transport cost and serves as the foundation for entropic regularization in the Sinkhorn iterations (Cuturi, 2013). Starting from all-ones scalings $\mathbf{u} \in \mathbb{R}^{m+1}$ and $\mathbf{v} \in \mathbb{R}^n$, we alternate between:

$$\mathbf{u}^{(r+1)} = \frac{[\boldsymbol{\mu}; s]}{\mathbf{K}\mathbf{v}^{(r)}}, \quad \mathbf{v}^{(r+1)} = \frac{\kappa\bar{\boldsymbol{\nu}}}{\mathbf{K}^{\top}\mathbf{u}^{(r+1)}}, \tag{6}$$

to satisfy the augmented marginals at each iteration $r$. After $T$ iterations, the transport plan is recovered as

$$\mathbf{\Pi}^{(T)} = \text{Diag}(\mathbf{u}^{(T)})\,\mathbf{K}\,\text{Diag}(\mathbf{v}^{(T)}). \tag{7}$$

We discard the dummy row and retain the first $m$ rows of $\mathbf{\Pi}^{(T)}$, denoted as $\mathbf{T}_{\text{real}}$, which satisfies $\mathbf{T}_{\text{real}}\mathbf{1}_n = \boldsymbol{\mu}$ and $\mathbf{T}_{\text{real}}^{\top}\mathbf{1}_m \leq \kappa\bar{\boldsymbol{\nu}}$. The Sinkhorn updates in Eq. (6)–(7) are efficiently computed in parallel over candidate subsets using batched matrix–vector operations. Finally, we compute the transport loss using only the real (non-dummy) mass:

$$\mathcal{L}_{\text{OT}} = \langle \mathbf{C}, \mathbf{T}_{\text{real}} \rangle = \sum_{i=1}^{m}\sum_{j=1}^{n} \mathbf{C}_{ij}\,\mathbf{T}_{\text{real},ij}, \tag{8}$$

where the entropy term in Eq. (5) serves internally to stabilize the Sinkhorn updates. This formulation enables flexible mass allocation, allowing selected samples to concentrate on representative regions while maintaining adequate coverage of the real distribution required for diffusion training.

### 3.4. Statistical Regularization

Although the one-sided partial OT cost effectively captures geometric alignment, it does not explicitly preserve distributional statistics or semantic clarity. We therefore introduce lightweight regularization terms that stabilize the condensation process and enhance distributional fidelity.

**Mean–variance regularization.** We align the first- and second-order statistics (mean and variance) of feature representations between the selected subset and the full dataset:

$$\mathcal{L}_{\text{sta}} = \|\boldsymbol{\mu}_{\mathcal{S}} - \boldsymbol{\mu}_{\mathcal{T}}\|_2^2 + \|\boldsymbol{\sigma}_{\mathcal{S}} - \boldsymbol{\sigma}_{\mathcal{T}}\|_2^2, \tag{9}$$

where $(\boldsymbol{\mu}_{\mathcal{S}}, \boldsymbol{\sigma}_{\mathcal{S}}^2)$ and $(\boldsymbol{\mu}_{\mathcal{T}}, \boldsymbol{\sigma}_{\mathcal{T}}^2)$ denote the mean and variance of features computed from $\mathcal{S}_{c(e)}$ and $\mathcal{T}_{c(e)}$, respectively. This regularization encourages the subset to approximate the global statistics of the real data distribution.

**Confidence regularization.** To ensure semantic consistency and avoid selecting samples that would act as unreliable geometric anchors, we introduce a lightweight confidence regularization term based on the predicted class probability:

$$\mathcal{L}_{\text{conf}} = \frac{1}{m}\sum_{i=1}^{m} -\log p(c|\mathbf{x}_i), \tag{10}$$

where $p(c|\mathbf{x}_i)$ denotes the predicted probability for the nominal class $c$. This term serves as a semantic reliability constraint during selection, discouraging low-confidence or off-class samples whose embeddings provide unreliable geometric evidence and would otherwise compromise the quality of geometry-preserving and distributional alignment.

The overall objective for each subset $\mathcal{S}_c$ is defined as:

$$\mathcal{L} = \mathcal{L}_{\text{OT}} + \alpha\mathcal{L}_{\text{sta}} + \beta\mathcal{L}_{\text{conf}}. \tag{11}$$

The weights $\alpha$ and $\beta$ balance the three loss terms.

### 3.5. Two-stage Discrete Optimization Strategy

Directly optimizing Eq. (11) over the discrete selection space

$$\mathcal{S}_c \subset \mathcal{T}_c, \quad |\mathcal{S}_c| = m \ll |\mathcal{T}_c|$$

is combinatorially intractable due to the exponential search space $\binom{|\mathcal{T}_c|}{m}$. We therefore adopt a two-stage optimization scheme integrating progressive coverage and refinement.

**Stage I: Greedy geometry-guided selection.** We construct the subset $\mathcal{S}_c$ in an incremental manner for initialization. Starting from an empty set $\mathcal{S}_c^{(0)} = \emptyset$, we evaluate the marginal gain at each step $t$ for adding each unselected candidate image $x_k \in \mathcal{T}_c \setminus \mathcal{S}_c^{(t-1)}$:

$$\Delta\mathcal{L}(x_k) = \mathcal{L}(\mathcal{S}_c^{(t-1)} \cup \{x_k\}, \mathcal{T}_c) - \mathcal{L}(\mathcal{S}_c^{(t-1)}, \mathcal{T}_c), \tag{12}$$

where $\mathcal{L}$ denotes the composite objective in Eq. 11. The sample that minimizes the incremental cost is selected:

$$x_t^* = \underset{x_k \in \mathcal{T}_c \setminus \mathcal{S}_c^{(t-1)}}{\arg\min} \ \Delta\mathcal{L}(x_k), \ \mathcal{S}_c^{(t)} = \mathcal{S}_c^{(t-1)} \cup \{x_t^*\}. \tag{13}$$

This greedy process iterates until $|\mathcal{S}_c^{(t)}| = t = m$. The objective $\mathcal{L}$ is efficiently computed in a batched manner.

**Stage II: Swap-based refinement.** While the greedy process provides a strong initialization, it may yield suboptimal early selections. We further refine the subset through pairwise swaps between selected and unselected samples. For each $x_i \in \mathcal{S}_c$ (iterated in fixed numerical order) and $x_j \in \mathcal{T}_c \setminus \mathcal{S}_c$ (evaluated in batched parallel form), we form a temporary subset $\mathcal{S}_c' = (\mathcal{S}_c \setminus \{x_i\}) \cup \{x_j\}$, and compute the relative improvement $\Delta_{i\to j}$:

$$\Delta_{i\to j} = \mathcal{L}(\mathcal{S}_c', \mathcal{T}_c) - \mathcal{L}(\mathcal{S}_c, \mathcal{T}_c). \tag{14}$$

*Table 1.* Comparison of FID-50K across data budgets using DiT-L/2 on ImageNet 256×256 (100k iters). Baselines: Random, K-Center (Gonzalez, 1985), Herding (Welling, 2009), and D$^2$C (Huang et al., 2026). Baseline results are taken from D$^2$C.

| Data Budget | Random | K-Center | Herding | D$^2$C | Ours |
|---|---|---|---|---|---|
| 0.8% (10K) | 35.86 | 50.77 | 40.75 | 4.20 | **3.43** |
| 4.0% (50K) | 36.78 | 69.86 | 32.38 | 14.81 | **11.01** |
| 8.0% (100K) | 41.02 | 71.31 | 36.37 | 22.55 | **17.09** |

If $\Delta_{i \to j} < 0$, the swap is accepted before moving to next $i$ (if multiple candidates $x_j$ satisfy the condition, we select the one that provides the largest improvement):

$$\mathcal{S}_c \leftarrow \mathcal{S}'_c, \tag{15}$$

and the refinement proceeds for a fixed number of rounds or until no further improvement is observed, i.e.,

$$\forall i, j, \quad \Delta_{i \to j} \geq 0. \tag{16}$$

This refinement alleviates greedy bias and emphasizes sample-wise contribution to the global geometric alignment and consistency. Together, the two stages provide an efficient discrete optimization procedure for our task, yielding compact subsets that remain faithful to the real distribution. Convergence analyses are provided in Appendix C. The pseudocode of the full method is provided in Appendix I.

## 4. Experiments

### 4.1. Experimental Settings

**Datasets.** We conduct experiments on ImageNet-1K (Russakovsky et al., 2015), training diffusion models using subsets of 10K, 50K, and 100K images, corresponding to about 0.8%, 4%, and 8% of the full dataset, respectively. All images are center-cropped and resized to 256×256 and 512×512 resolutions following the ADM preprocessing pipeline (Dhariwal & Nichol, 2021).

**Networks.** Following D$^2$C (Huang et al., 2026), we adopt two diffusion variants: DiT-L/2 (Peebles & Xie, 2023) and SiT-L/2 (Ma et al., 2024). Both models are trained from scratch on the selected subsets.

**Baselines.** We compare our method against a wide range of real subset selection baselines, including Herding (Welling, 2009), K-Center (Gonzalez, 1985; Sener & Savarese, 2018), Random Sampling, Random† (with the attach phase), CCS (Zheng et al., 2023), DQ (Zhou et al., 2023), and D$^2$C (Huang et al., 2026). For fairness, Random†, CCS, DQ, D$^2$C, and our method all employ the same attach phase implementation as D$^2$C (Huang et al., 2026), where each selected image is enriched with textual and visual embeddings during diffusion model training. Synthetic set generation baselines are compared in Appendix G.2.

*Table 2.* Comparison across different methods using DiT-L/2 (Peebles & Xie, 2023) on ImageNet 256×256 with 100K iterations and 10K data budget. We use 50K generated samples for evaluation. Random† adopts the attach phase from D$^2$C (Huang et al., 2026).

| Method | Data Budget | FID↓ | IS↑ | Precision↑ | Recall↑ |
|---|---|---|---|---|---|
| Random† | 0.8% (10K) | 4.63 | 263.1 | 0.70 | 0.26 |
| CCS (Zheng et al., 2023) | 0.8% (10K) | 5.45 | 364.9 | 0.77 | 0.21 |
| DQ (Zhou et al., 2023) | 0.8% (10K) | 4.56 | 267.8 | 0.72 | 0.25 |
| D$^2$C (Huang et al., 2026) | 0.8% (10K) | 4.20 | 283.6 | 0.72 | 0.24 |
| Ours | 0.8% (10K) | **3.43** | **414.3** | **0.78** | **0.28** |

**Evaluation Metrics.** Following prior works (Huang et al., 2026; Peebles & Xie, 2023), we evaluate the generative performance using Fréchet Inception Distance (FID) (Heusel et al., 2017), Inception Score (IS) (Salimans et al., 2016), Precision, and Recall metrics. FID measures image fidelity and diversity, IS reflects semantic clarity, while Precision and Recall jointly assess generation fidelity and coverage.

The implementation details are provided in Appendix E.

### 4.2. Results and Discussions

**Overall Comparison on ImageNet 256×256.** We first evaluate our method on ImageNet (Russakovsky et al., 2015) 256 × 256 within the DiT-L/2 (Peebles & Xie, 2023) setup. As shown in Tables 1, 2, and 3, our method consistently achieves the lowest FID and highest IS across all data budgets, outperforming classical selection strategies such as Random, K-Center (Sener & Savarese, 2018), and Herding (Welling, 2009), as well as recent selection approaches like CCS (Zheng et al., 2023), DQ (Zhou et al., 2023) and D$^2$C (Huang et al., 2026). The improvement arises from two complementary mechanisms: (1) OT establishes explicit geometric alignment between the full and selected sets, while its one-sided partial form adaptively reallocates transport mass to emphasize informative regions; and (2) statistical alignment regularization enforces both global moment consistency (mean and variance) and implicit semantic agreement between the selected subset and the full dataset, stabilizing condensation and preserving balanced class semantics. Together, these mechanisms yield subsets that are both geometrically faithful and statistically representative, enabling diffusion models to achieve high generative fidelity using only a small fraction of the original data.

Here we fix the total number of training iterations at 100K to ensure a fair comparison between different condensation methods under the same training conditions. The purpose here IS not to compare different data budgets, but to evaluate which condensation method performs best in terms of FID and other metrics within the same number of iterations. By fixing the iteration count, we ensured consistency across methods, making the comparison between different condensation methods clear and direct. Data budget refers to the size of the subset that is selected for efficient training. The

*Table 3.* Comparison across different methods and data budgets using DiT-L/2 (Peebles & Xie, 2023) on ImageNet 256×256 with 100K iterations. We use 50K generated samples for evaluation.

| Method | Data Budget | FID↓ | IS↑ | Precision↑ | Recall↑ |
|---|---|---|---|---|---|
| Random† | 4.0% (50K) | 14.00 | 120.1 | 0.67 | 0.53 |
| CCS (Zheng et al., 2023) | 4.0% (50K) | 15.21 | 150.4 | 0.67 | 0.46 |
| DQ (Zhou et al., 2023) | 4.0% (50K) | 13.56 | 123.3 | 0.67 | 0.52 |
| D²C (Huang et al., 2026) | 4.0% (50K) | 14.81 | 131.8 | **0.68** | 0.50 |
| Ours | 4.0% (50K) | **11.01** | **174.2** | **0.68** | **0.57** |
| Random† | 8.0% (100K) | 19.80 | 90.6 | 0.66 | 0.54 |
| CCS (Zheng et al., 2023) | 8.0% (100K) | 22.36 | 101.5 | 0.66 | 0.48 |
| DQ (Zhou et al., 2023) | 8.0% (100K) | 20.16 | 84.5 | 0.64 | 0.54 |
| D²C (Huang et al., 2026) | 8.0% (100K) | 22.55 | 96.7 | 0.66 | 0.53 |
| Ours | 8.0% (100K) | **17.09** | **116.3** | **0.67** | **0.59** |

*Table 4.* Comparison across different methods using DiT-L/2 (Peebles & Xie, 2023) on ImageNet (Russakovsky et al., 2015) 512×512 with 100K iterations and 10K data budget. Baseline results are taken from D²C (Huang et al., 2026).

| Method | Data Budget | FID↓ | IS↑ | Precision↑ | Recall↑ |
|---|---|---|---|---|---|
| Random | 0.8% (10K) | 24.8 | 74.3 | 0.65 | 0.42 |
| D²C (Huang et al., 2026) | 0.8% (10K) | 14.8 | 109.2 | 0.63 | 0.52 |
| Ours | 0.8% (10K) | **6.17** | **451.0** | **0.81** | **0.67** |

*Table 5.* Comparison of FID-50K across data budgets using SiT-L/2 on ImageNet 256×256 (100k iters). Baselines: Random, K-Center (Gonzalez, 1985), Herding (Welling, 2009), and D²C (Huang et al., 2026). Baseline results are taken from D²C.

| Data Budget | Random | K-Center | Herding | D²C | Ours |
|---|---|---|---|---|---|
| 0.8% (10K) | 4.35 | 14.77 | 22.96 | 3.98 | **3.56** |
| 4.0% (50K) | 31.13 | 61.66 | 29.11 | 11.21 | **7.26** |
| 8.0% (100K) | 36.64 | 66.96 | 32.30 | 15.01 | **8.83** |

*Table 6.* Comparison across different methods using SiT-L/2 (Ma et al., 2024) on ImageNet 256×256 (100K training iterations and 100K data budget). We use 50K generated samples for evaluation. Random† adopts the attach phase from D²C (Huang et al., 2026).

| Method | Data Budget | FID↓ | IS↑ | Precision↑ | Recall↑ |
|---|---|---|---|---|---|
| Random† | 8.0% (100K) | 15.58 | 125.9 | 0.73 | 0.50 |
| CCS (Zheng et al., 2023) | 8.0% (100K) | 15.41 | 142.9 | **0.74** | 0.43 |
| DQ (Zhou et al., 2023) | 8.0% (100K) | 15.24 | 121.9 | 0.73 | 0.50 |
| D²C (Huang et al., 2026) | 8.0% (100K) | 15.01 | 127.6 | 0.73 | 0.48 |
| Ours | 8.0% (100K) | **8.83** | **167.5** | **0.74** | **0.53** |

Table 6, our method continues to achieve consistent improvements over all baselines under the SiT setting, complementing the DiT results reported earlier. This demonstrates that the selected subsets preserve transferable data-level statistics that generalize across different diffusion formulations, showing that our condensation framework provides a reliable data substrate for diffusion-style generative training.

**Evaluation under Different Protocols.** We further validate our approach under multiple evaluation protocols. Table 7 reports quantitative results using 10K generated samples to complement the standard 50K-sample evaluation in Tables 2 and 3. In addition, Table 8 investigates the effect of extended training iterations (up to 300K). Across all evaluation settings, our method consistently achieves the best results, indicating that the improvements are intrinsic and not tied to a specific evaluation configuration. Moreover, Figure 3 shows that our method reaches a substantially lower FID plateau compared with prior approaches, demonstrating that the selected subsets lead to more stable and faithful generative training. Overall, these results confirm that our condensation strategy produces robust and high-quality subsets whose advantages persist across evaluation scales.

**Ablation Studies.** We conduct ablation studies to evaluate

observed trend is a result of fixed-iteration convergence lag. With a constant batch size and iteration count (100K), a larger data budget (e.g., 100K vs 10K images) drastically reduces the number of "epochs" or updates per sample. Specifically, for a 10K subset, each sample is seen ∼1280 times, whereas for a 100K subset, it is seen only ∼128 times. Although larger budgets provide a higher performance ceiling, they require significantly more iterations to reach it. In the early stages of training, with the same number of iterations, smaller data budgets can perform better if the selection is well-optimized, as each sample is updated more frequently. This confirms our method's core advantage: accelerating convergence by focusing the training process on a geometry-consistent core manifold.

**Results on Higher Image Resolution.** We further evaluate the advantages of our selected set at 512 × 512 resolution. As shown in Table 4, our method continues to outperform both random and D²C selections, achieving substantial improvements in FID and IS. This demonstrates that our strategy generalizes robustly across resolutions: even when the input resolution increases, the same selected subset still provides representative coverage of high-frequency details and global structures. Such resolution robustness suggests that our method captures intrinsic, scale-consistent data semantics rather than memorizing low-resolution appearance patterns. More results are provided in Appendix G.1.

**Robustness across Diffusion Variants.** We further evaluate our condensation strategy on SiT-L/2 (Ma et al., 2024), an interpolant-based diffusion variant of DiT-L/2 (Peebles & Xie, 2023) that replaces the standard noise-prediction objective with a velocity-based interpolant formulation bridging diffusion and flow dynamics. As shown in Table 5 and

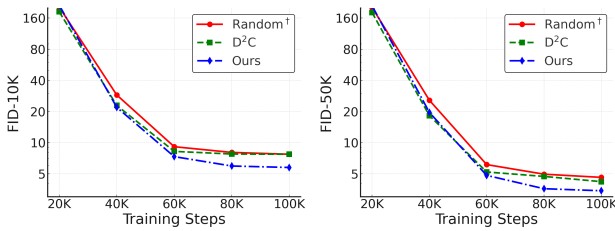

*Figure 3.* FID comparison between Random†, D²C (Huang et al., 2026), and Ours across different training steps, where the left and right panels respectively show FID-10K and FID-50K.

*Table 7.* Comparison across different methods and data budgets using DiT-L/2 (Peebles & Xie, 2023) on ImageNet 256×256 with 100K iterations. We use 10K generated samples for evaluation. Random[†] adopts the attach phase from D²C (Huang et al., 2026).

| Method | Data Budget | FID↓ | IS↑ | Precision↑ | Recall↑ |
|---|---|---|---|---|---|
| Random[†] | 0.8% (10K) | 7.73 | 263.1 | 0.69 | 0.68 |
| CCS (Zheng et al., 2023) | 0.8% (10K) | 7.81 | 373.3 | 0.77 | 0.58 |
| DQ (Zhou et al., 2023) | 0.8% (10K) | 7.55 | 271.0 | 0.73 | **0.69** |
| D²C (Huang et al., 2026) | 0.8% (10K) | 7.73 | 279.8 | 0.72 | **0.69** |
| Ours | 0.8% (10K) | **5.76** | **430.2** | **0.78** | **0.69** |
| Random[†] | 4.0% (50K) | 17.04 | 120.8 | 0.67 | 0.65 |
| CCS (Zheng et al., 2023) | 4.0% (50K) | 17.21 | 147.9 | 0.67 | 0.60 |
| DQ (Zhou et al., 2023) | 4.0% (50K) | 16.83 | 122.8 | 0.67 | 0.65 |
| D²C (Huang et al., 2026) | 4.0% (50K) | 16.28 | 131.1 | **0.68** | 0.63 |
| Ours | 4.0% (50K) | **13.73** | **173.5** | **0.68** | **0.69** |

*Table 8.* Comparison across different methods using DiT-L/2 (Peebles & Xie, 2023) on ImageNet 256×256 with 300K iterations and 50K data budget. We use 50K generated samples for evaluation. Random[†] adopts the attach phase from D²C (Huang et al., 2026).

| Method | Data Budget | FID↓ | IS↑ | Precision↑ | Recall↑ |
|---|---|---|---|---|---|
| Random[†] | 4.0% (50K) | 6.02 | 188.5 | 0.69 | 0.58 |
| CCS (Zheng et al., 2023) | 4.0% (50K) | 5.97 | 259.3 | 0.70 | 0.51 |
| DQ (Zhou et al., 2023) | 4.0% (50K) | 4.33 | 214.2 | 0.70 | 0.58 |
| D²C (Huang et al., 2026) | 4.0% (50K) | 4.76 | 207.0 | 0.70 | 0.57 |
| Ours | 4.0% (50K) | **3.34** | **327.7** | **0.71** | **0.61** |

the contribution of each component. As shown in Table 9, removing $\mathcal{L}_{\mathrm{OT}}$ degrades FID and reduces both precision and recall, indicating that the geometric constraint is essential for promoting coverage and faithful geometric alignment between the condensed and real distributions. Replacing the one-sided partial OT with the classical balanced OT degrades performance, indicating that relaxing alignment on peripheral samples yields more coherent and geometry-consistent subsets under small budgets. Further analysis of partial transport behavior is provided in Appendix G.4. Eliminating $\mathcal{L}_{\mathrm{sta}}$ greatly worsens FID, verifying that enforcing global distribution prevents mode imbalance and stabilizes training. Removing $\mathcal{L}_{\mathrm{conf}}$ primarily decreases IS, demonstrating that the semantic constraint improves class-level clarity by filtering uncertain or low-confidence samples. More discussion on confidence regularization can be found in Appendix G.6. Table 10 further evaluates Stage II refinement. Introducing this stage consistently improves FID, precision, and recall compared with its absence, confirming that iterative exchange refines subset geometry, strengthens manifold coverage, and preserves compactness.

*Table 9.* Ablation study of key components in our dataset condensation framework, conducted on ImageNet at 256×256 resolution for 100K training iterations with a 10K-sample data budget.

| Method | FID↓ | IS↑ | Precision↑ | Recall↑ |
|---|---|---|---|---|
| w/o $\mathcal{L}_{\mathrm{OT}}$ | 3.82 | 414.1 | 0.72 | 0.26 |
| w/o $\mathcal{L}_{\mathrm{sta}}$ | 4.62 | 451.6 | 0.81 | 0.26 |
| w/o $\mathcal{L}_{\mathrm{conf}}$ | 3.55 | 337.0 | 0.79 | 0.27 |
| balanced OT (w/o partial) | 3.54 | 413.9 | 0.75 | 0.28 |
| Ours (full) | 3.43 | 414.3 | 0.78 | 0.28 |

*Table 10.* Ablation study on Stage II optimization (swap-based refinement) on ImageNet 256×256 for 100K iterations.

| Method | Data Budget | FID↓ | IS↑ | Precision↑ | Recall↑ |
|---|---|---|---|---|---|
| w/o Stage II | 0.8% (10K) | 3.82 | **417.6** | 0.74 | 0.25 |
| w Stage II | 0.8% (10K) | **3.43** | 414.3 | **0.78** | **0.28** |
| w/o Stage II | 4.0% (50K) | 12.87 | 172.4 | 0.61 | **0.57** |
| w Stage II | 4.0% (50K) | **11.01** | **174.2** | **0.68** | **0.57** |

*Table 11.* Impact of $\kappa$ (capacity scaling) and $\gamma$ (dummy-source) on ImageNet 256×256 (10K data budget, 100K iterations).

| $\kappa$ | 1.01 | 1.03 | 1.05 | 1.07 | 1.10 |
|---|---|---|---|---|---|
| FID-50K | 3.49 | 3.45 | 3.43 | 3.46 | 3.58 |
| $\gamma$ | 0.3 | 0.5 | 0.7 | 1.0 | 2.0 |
| FID-50K | 3.43 | 3.43 | 3.43 | 3.46 | 3.47 |

**Sensitivity Analysis.** Table 11 shows the effect of varying the capacity scaling factor $\kappa$ and the dummy-source parameter $\gamma$, which determines the dummy-source cost $\delta$. The results indicate that FID remains stable across a wide range of settings, demonstrating low sensitivity and robust optimization behavior. However, an excessively large $\kappa$ may degrade performance, as discarding too many samples in partial OT weakens the retained information content. Analyses on $\alpha, \beta, \varepsilon, T$ are provided in Appendix F.

**Visualizations.** Figures 1 and 4 respectively show the selected sets and the images generated by DiT models trained on them. The subset produced by D²C (Huang et al., 2026) sometimes contains visually homogeneous exemplars with consistent style, and in other cases includes semantically ambiguous selections that do not clearly represent the class. As a result, DiT trained on this subset exhibits reduced coverage of intra-class variation and may produce samples that either collapse to a few modes or drift semantically. In contrast, our selected set provides broader manifold coverage and clearer exemplars, enabling the DiT to synthesize images with richer diversity and faithful class characteristics. See Appendix G.4 and H for more visualizations.

**Runtime Analysis.** We measure the selection runtime on ImageNet (Russakovsky et al., 2015) with a 10K data budget. As summarized in Table 12, our method substantially

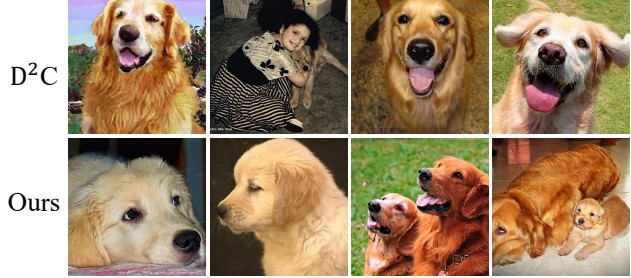

*Figure 4.* Visualization of generated images (golden retriever).

*Table 12.* Real subset selection time on ImageNet with a 10K data budget, measured on NVIDIA RTX 3090 GPUs.

| Number of GPUs | DQ (Zhou et al., 2023) | D²C (Huang et al., 2026) | Ours |
|---|---|---|---|
| 1 | 30.4 hours | 41.9 hours | **5.5 hours** |
| 8 | 238 mins | 314 mins | **96 mins** |

reduces wall-clock time compared with DQ (Zhou et al., 2023) and D²C (Huang et al., 2026). The detailed runtime breakdown in Table 13 shows that data loading and preprocessing dominates computation (about 65%), while greedy initialization and swap refinement together account for the remaining time. This demonstrates that the discrete optimization stages are lightweight and practical in use.

*Table 13.* Runtime breakdown of our pipeline on ImageNet with a 10K data budget, measured on 8×RTX 3090 GPUs.

| Component | Data & Feature | Stage I (Greedy Init) | Stage II (Swap Refine) |
|---|---|---|---|
| Percentage | 65.1% | 9.0% | 25.9% |

## 5. Conclusion

This work presents a geometry-aware alignment-driven framework for dataset condensation in diffusion model training. We formulate the problem as a constrained discrete optimization task, where a one-sided POT objective selectively aligns informative and stable regions of the data manifold. By regularizing feature moments and class consistency, we further improve statistical and semantic coherence. The proposed two-stage optimization, consisting of geometry-guided greedy selection and swap-based refinement, efficiently approximates the global optimum with low computational cost. Experiments across different diffusion variants, subset sizes, training steps, and evaluation settings demonstrate consistent improvements in diffusion fidelity.

## Impact Statement

This paper presents work whose goal is to improve the data efficiency of diffusion model training through dataset condensation. By reducing the amount of training data required, the proposed approach may help lower computational cost and energy consumption, supporting more efficient use of large-scale generative models. As with many data reduction and generative modeling techniques, applying dataset condensation may alter the underlying data distribution and requires careful consideration in practice. Potential risks related to bias, privacy, and misuse are not unique to this work and can be addressed through established practices such as responsible dataset curation, evaluation, and transparent reporting. Overall, this work advances research in data-efficient generative modeling. We do not foresee any novel or uniquely harmful societal impacts beyond those already associated with diffusion models and dataset condensation methods more broadly. A more detailed discussion of broader impacts is provided in Appendix J.

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

# Appendix of Geometry-Aware Dataset Condensation for Diffusion Model Training

**Overview** This appendix provides extended technical details and supplementary analyses to complement the main text. We begin with additional related works on optimal transport in **Section A**, followed by a rigorous dummy-source reformulation together with entropic Sinkhorn validation in **Section B**. **Section C** presents the theoretical convergence analysis of our two-stage selection framework, while **Section D** compiles all mathematical symbols used throughout the paper. Implementation details and additional hyperparameter sensitivity studies are provided in **Section E** and **Section F**, respectively. Further experimental analyses are collected in **Section G**, including complementary $512 \times 512$ experiments in **Section G.1**, comparisons with additional baselines in **Section G.2**, ablations on the feature encoder and classifier in **Section G.3**, an in-depth examination of one-sided partial optimal transport in **Section G.4**, the adaptation of existing distribution-matching objectives within our framework in **Section G.5**, clarification for the meaning of "geometry" in **Section G.6**, and more discussion on confidence regularization in **Section G.7**. We then present additional visualization results in **Section H**, provide the full pseudocode of our method in **Section I**, which includes the end-to-end condensation pipeline and its core subroutines, and conclude with a discussion of broader impacts in **Section J**.

> **Key Highlights**
>
> 1. Our method consistently outperforms existing real subset selection methods and synthetic set generation approaches in dataset condensation for diffusion model training. (see Section G.2).
> 2. Our method is feature-encoder and classifier agnostic (see Section G.3).
> 3. We provide an analysis of the two-stage discrete optimization scheme, including upper-bound characterizations under mild assumptions (see Appendix C).
> 4. Our framework remains stable across broad hyperparameter ranges, greatly simplifying tuning (see Section F).
> 5. Our one-sided partial OT suppresses peripheral low-density samples that would otherwise distort distance geometry and pull the condensed set away from the core manifold, preventing biased global and inconsistent local alignment (see Section G.4).
> 6. Our alignment-driven selection framework is compatible with some existing distribution-matching-based synthetic set generation objectives and allows them to be seamlessly adapted to discrete real subset selection (see Section G.5).

## A. Related Works on Optimal Transport

Optimal Transport (OT) provides a principled geometric framework for comparing probability distributions by computing the minimal cost required to transform one distribution into another. Compared to divergences such as KL and Jensen–Shannon, OT remains meaningful even when supports do not overlap and thus yields a more faithful notion of distributional discrepancy (Villani & Villani, 2009; Zhang et al., 2021). The resulting Wasserstein distance has been widely adopted in image or mesh generation (Arjovsky et al., 2017; Gulrajani et al., 2017; Sun et al., 2026), causal discovery (Wheeler & Natarajan, 2025; Chen et al., 2023), unsupervised learning (Luo et al., 2025; Liu et al., 2025; Montesuma et al., 2025), and reinforcement learning (Baheri et al., 2025; Klink et al., 2024; Lan et al., 2023; Zhang et al., 2024b), but its exact computation is prohibitive in high dimensions. Entropy-regularized OT, or Sinkhorn distance (Cuturi, 2013), alleviates this limitation and has enabled scalable OT-based methods in domain adaptation (Lyu et al., 2025; Zeng et al., 2024; Koç et al., 2025), classification (Nguyen et al., 2025; Jin et al., 2025), and knowledge distillation (Cui et al., 2025c; 2024a;b).

OT has also been exploited to improve diffusion models. AOT (Kim et al., 2024) reduces the curvature of the ODE trajectories and truncation errors during sampling via Approximated OT, leading to fewer sampling steps and improved stability. DPM-OT (Li et al., 2023) instead applies OT at the sampling stage, learning a direct mapping from noise to data that substantially shortens the reverse diffusion trajectory. Both AOT and DPM-OT apply OT on the model side to reshape the generative dynamics, which primarily affects the sampling process. In contrast, our approach adopts a data-centric perspective and uses OT to decide which real samples should be included during training.

Partial optimal transport (POT) extends classical OT by transporting only a fixed amount of mass, inducing an active region

with a free boundary. Caffarelli and McCann (Caffarelli & McCann, 2010) and Figalli (Figalli, 2010) established existence, uniqueness, and regularity for this problem, while Séjourné et al. (Séjourné et al., 2023) positioned POT within the broader theory of generalized and unbalanced OT. Algorithmically, Chapel et al. (Chapel et al., 2020) leveraged POT for robust positive–unlabeled learning, and Bai et al. developed scalable variants via Sliced Optimal Partial Transport (Bai et al., 2023b) and Linear Optimal Partial Transport Embedding (Bai et al., 2023a). POT has further supported representation alignment in universal domain adaptation (Yang et al., 2023) and open-set OOD detection (Ren et al., 2024).

Most existing POT formulations relax mass constraints on *both* sides to allow bidirectional partial matching. In contrast, our task imposes an inherently asymmetric structure: the condensed set must fully participate in matching, while only a subset of the full dataset should be used. We therefore adopt a *one-sided* partial OT formulation tailored to this geometry, and design a discrete optimization scheme that balances convergence stability with practical selection efficiency.

## B. Dummy-source Reformulation and Validation

Recall Eq. (3) in the main text defines a one-sided partial OT where the source mass is fully transported while the target side is relaxed by a capacity factor $\kappa \geq 1$:

$$\min_{\boldsymbol{\pi} \geq 0} \langle \mathbf{C}, \boldsymbol{\pi} \rangle \quad \text{s.t.} \quad \boldsymbol{\pi} \mathbf{1}_n = \boldsymbol{\mu}, \ \boldsymbol{\pi}^\top \mathbf{1}_m \leq \kappa \bar{\boldsymbol{\nu}}. \tag{17}$$

Here $\mathbf{C} \in \mathbb{R}^{m \times n}$ is the pairwise cost matrix, and $\boldsymbol{\mu} = \frac{1}{m} \mathbf{1}_m, \boldsymbol{\nu} = \frac{1}{n} \mathbf{1}_n$ denote the reference marginals on the source and target sides, respectively. The relaxation factor $\kappa$ allows a portion of the target mass to remain unmatched.

**Step 1: Introduce slack on the target side.** Define the unused target capacity as a nonnegative slack vector:

$$\mathbf{r} \triangleq \kappa \bar{\boldsymbol{\nu}} - \boldsymbol{\pi}^\top \mathbf{1}_m \in \mathbb{R}_+^n. \tag{18}$$

Left-multiplying $\mathbf{1}_n^\top$ and using $\boldsymbol{\pi} \mathbf{1}_n = \boldsymbol{\mu}$ yields

$$\mathbf{1}_n^\top \mathbf{r} = \kappa \mathbf{1}_n^\top \bar{\boldsymbol{\nu}} - \mathbf{1}_m^\top \boldsymbol{\pi} \mathbf{1}_n = \kappa - 1 \triangleq s. \tag{19}$$

Hence, the total slack corresponds to a dummy supply $s = \kappa - 1$.

**Step 2: Augment the transport plan and marginals.** Stack $\mathbf{r}^\top$ under $\boldsymbol{\pi}$ to form an augmented plan:

$$\boldsymbol{\Pi} = \begin{bmatrix} \boldsymbol{\pi} \\ \mathbf{r}^\top \end{bmatrix} \in \mathbb{R}_+^{(m+1) \times n}, \qquad \tilde{\boldsymbol{\mu}} = \begin{bmatrix} \boldsymbol{\mu} \\ s \end{bmatrix}, \qquad \tilde{\boldsymbol{\nu}} = \kappa \bar{\boldsymbol{\nu}}. \tag{20}$$

Then the balanced marginal constraints become

$$\boldsymbol{\Pi} \mathbf{1}_n = \tilde{\boldsymbol{\mu}}, \qquad \boldsymbol{\Pi}^\top \mathbf{1}_{m+1} = \tilde{\boldsymbol{\nu}}, \tag{21}$$

which are algebraically equivalent to those in Eq. (3): the second equality implies $\boldsymbol{\pi}^\top \mathbf{1}_m = \kappa \bar{\boldsymbol{\nu}} - \mathbf{r} \leq \kappa \bar{\boldsymbol{\nu}}$.

**Step 3: Augment the cost matrix.** To absorb unmatched target mass, append a dummy-source row with constant cost $\delta > 0$:

$$\mathbf{C}_{\text{aug}} = \begin{bmatrix} \mathbf{C} \\ \delta \mathbf{1}_n^\top \end{bmatrix}. \tag{22}$$

The objective becomes

$$\langle \mathbf{C}_{\text{aug}}, \boldsymbol{\Pi} \rangle = \langle \mathbf{C}, \boldsymbol{\pi} \rangle + \delta \mathbf{1}_n^\top \mathbf{r} = \langle \mathbf{C}, \boldsymbol{\pi} \rangle + \delta s, \tag{23}$$

where the additional term $\delta s$ is constant w.r.t. $\boldsymbol{\pi}$. Thus, minimizing Eq. (3) is equivalent to minimizing:

$$\min_{\boldsymbol{\Pi} \geq 0} \langle \mathbf{C}_{\text{aug}}, \boldsymbol{\Pi} \rangle \quad \text{s.t.} \quad \boldsymbol{\Pi} \mathbf{1}_n = \tilde{\boldsymbol{\mu}}, \ \boldsymbol{\Pi}^\top \mathbf{1}_{m+1} = \tilde{\boldsymbol{\nu}}. \tag{24}$$

Eqs. (18)–(24) show that the feasible sets of Eq. (17) and Eq. (24) are in one-to-one correspondence, and the objectives differ by a constant $\delta s$. Hence both problems share identical minimizers for the real transport plan $\boldsymbol{\pi}$.

**Step 4: Entropic regularization and Sinkhorn formulation.** For differentiability and computational efficiency, we add an entropy term $H(\mathbf{\Pi}) = -\sum_{ij} \mathbf{\Pi}_{ij}(\log \mathbf{\Pi}_{ij} - 1)$ with weight $\varepsilon > 0$, yielding the entropic OT:

$$\min_{\mathbf{\Pi} \geq 0} \langle \mathbf{C}_{\mathrm{aug}}, \mathbf{\Pi} \rangle - \varepsilon H(\mathbf{\Pi}) \quad \text{s.t.} \quad \mathbf{\Pi} \mathbf{1}_n = \tilde{\boldsymbol{\mu}}, \ \mathbf{\Pi}^\top \mathbf{1}_{m+1} = \tilde{\boldsymbol{\nu}}, \tag{25}$$

which corresponds exactly to Eq. (5) in the main text. The optimal solution admits the Gibbs–Sinkhorn form:

$$\mathbf{\Pi}^\varepsilon = \mathrm{diag}(\mathbf{u}) \, \mathbf{K} \, \mathrm{diag}(\mathbf{v}), \qquad \mathbf{K} = \exp\left(-\frac{\mathbf{C}_{\mathrm{aug}}}{\varepsilon}\right), \tag{26}$$

where $(\mathbf{u}, \mathbf{v})$ are the scaling vectors enforcing $\tilde{\boldsymbol{\mu}}, \tilde{\boldsymbol{\nu}}$. As $\varepsilon \to 0$, the entropic solution $\mathbf{\Pi}^\varepsilon$ converges to the linear programming optimum of Eq. (24), thus recovering the original partial OT formulation.

*On the role and effect of $\delta$.* Even with $\delta = 0$, the dummy row can only carry a fixed total mass $s = \kappa - 1$. However, when $\varepsilon$ is small and $\mathbf{C}$ spans a wide range, $K_{m+1,j} = e^{-\delta/\varepsilon}$ with $\delta = 0$ may be much larger than many real entries, leading to ill-conditioned scaling (extreme $u, v$). An excessively large $\delta$ makes the dummy row numerically inactive ($K_{m+1,j} \approx 0$) yet still required to carry a fixed total mass $s$, forcing the algorithm to balance an unrealistically high-cost assignment and potentially destabilizing the Sinkhorn updates. Hence $\delta$ should remain within the typical scale of the transport costs.

**Remarks.** (i) When $\kappa = 1$, the dummy source vanishes ($s = 0$) and Eq. (25) reduces to balanced OT. (ii) The dummy-source cost $\delta$ is a positive constant ensuring numerical stability; in practice, $\delta = \gamma \cdot \mathrm{median}(\mathbf{C})$. (iii) The last row of $\mathbf{\Pi}$ corresponds solely to the dummy source, while the first $m$ rows represent the real transport plan used in our loss computation.

## C. Convergence Analysis

**Setup and notation.** Fix a class $c$ with ground set $\mathcal{T}_c$ (all candidate real samples) and target subset size $m \ll |\mathcal{T}_c|$. Let $S_c^{(t)} \subset \mathcal{T}_c$ denote the subset after $t$ greedy additions, with $|S_c^{(t)}| = t$ and $S_c^{(0)} = \varnothing$. Our composite loss is $\mathcal{L}(S_c, \mathcal{T}_c)$ and we define the utility

$$F(S_c) \triangleq -\mathcal{L}(S_c, \mathcal{T}_c). \tag{27}$$

Larger $F$ means better performance. Let $S_c^\star \in \arg\max_{|S| \leq m} F(S)$.

For any $A \subseteq B \subseteq \mathcal{T}_c$ and $X \subseteq \mathcal{T}_c \setminus B$, the *weak submodularity ratio* (Das & Kempe, 2011; Elenberg et al., 2016) is

$$\gamma_{\mathrm{wm}} := \inf_{A \subseteq B, \ X \cap B = \varnothing} \frac{\sum_{x \in X} \left[ F(B \cup \{x\}) - F(B) \right]}{F(B \cup X) - F(B)} \ \in \ (0, 1]. \tag{28}$$

### C.1. Stage I (Greedy Geometry-guided Selection).

At iteration $t$, add $x_t^\star \in \mathcal{T}_c \setminus S_c^{(t-1)}$ maximizing $F(S_c^{(t-1)} \cup \{x\}) - F(S_c^{(t-1)})$. Under weak submodularity (Das & Kempe, 2011; Elenberg et al., 2016), for $t = 1, \ldots, m$,

$$F(S_c^{(t)}) - F(S_c^{(t-1)}) \ \geq \ \frac{\gamma_{\mathrm{wm}}}{m} \left( F(S_c^\star) - F(S_c^{(t-1)}) \right). \tag{29}$$

Let $\Delta_t := F(S_c^\star) - F(S_c^{(t)})$ be the optimality gap. Then

$$\Delta_t \ \leq \ \left( 1 - \frac{\gamma_{\mathrm{wm}}}{m} \right) \Delta_{t-1}, \qquad t = 1, \ldots, m. \tag{30}$$

**Gap bound relative to $S_c^{(1)}$.** Unrolling (30) from $t = 2$ gives, for $t \geq 2$,

$$\Delta_t \ \leq \ \left( 1 - \frac{\gamma_{\mathrm{wm}}}{m} \right)^{t-1} \Delta_1. \tag{31}$$

Equivalently,

$$F(S_c^{(t)}) \ \geq \ F(S_c^\star) - \left( 1 - \frac{\gamma_{\mathrm{wm}}}{m} \right)^{t-1} \left( F(S_c^\star) - F(S_c^{(1)}) \right), \quad t = 2, \ldots, m, \tag{32}$$

or in loss form,

$$\mathcal{L}(S_c^{(t)}, \mathcal{T}_c) - \mathcal{L}(S_c^{\star}, \mathcal{T}_c) \ \leq \ \left(1 - \frac{\gamma_{\mathrm{wm}}}{m}\right)^{t-1} \Big[\mathcal{L}(S_c^{(1)}, \mathcal{T}_c) - \mathcal{L}(S_c^{\star}, \mathcal{T}_c)\Big], \quad t = 2, \ldots, m. \tag{33}$$

At $t = m$,

$$\boxed{\mathcal{L}(S_c^{(m)}, \mathcal{T}_c) - \mathcal{L}(S_c^{\star}, \mathcal{T}_c) \ \leq \ \left(1 - \frac{\gamma_{\mathrm{wm}}}{m}\right)^{m-1} \Big[\mathcal{L}(S_c^{(1)}, \mathcal{T}_c) - \mathcal{L}(S_c^{\star}, \mathcal{T}_c)\Big].} \tag{34}$$

This quantifies the contraction of the residual gap after the first greedy addition.

### C.2. Stage II (Swap-based Refinement)

**Assumptions for Stage II.** Throughout Stage II we assume a *restricted monotonicity* (RM) condition in size-$m$ neighborhoods. Let $\mathcal{F}_m := \{S \subseteq \mathcal{T}_c : |S| = m\}$.

**Restricted monotonicity (RM) at size-$m$ contexts.** We say $F$ satisfies RM if for any $U \in \mathcal{F}_m$ and any $X \subseteq \mathcal{T}_c \setminus U$ with $|X| \leq m$,

$$F(U \cup X) \ \geq \ F(U). \tag{35}$$

This is weaker than global monotonicity and is the only monotonicity-type condition required in Lemma C.2.

*Restricted block–singleton curvature.* We say $F$ has curvature (Iyer et al., 2013; Conforti & Cornuéjols, 1984; Vondrák) $\kappa_b \in [0, 1)$ on $\mathcal{F}_m$ if

$$\sum_{v \in V} \big(F(U) - F(U \setminus \{v\})\big) \ \leq \ \frac{1}{1 - \kappa_b} \big(F(U) - F(U \setminus V)\big), \qquad \forall U \in \mathcal{F}_m, \ \forall V \subseteq U. \tag{36}$$

This depends only on differences of $F$ and is invariant under constant shifts.

*Restricted approximate diminishing-returns (DR) ratio.* We say $F$ admits $\alpha_{\mathrm{DR}} \in (0, 1]$ at size-$m$ contexts if

$$F(T \cup \{x\}) - F(T) \ \geq \ \alpha_{\mathrm{DR}}\big(F(U \cup \{x\}) - F(U)\big), \quad \forall U \in \mathcal{F}_m, \ \forall T \subseteq U, \ \forall x \notin U. \tag{37}$$

When $\alpha_{\mathrm{DR}} = 1$, this is exact DR (submodularity).

**Stage II setup.** Let $S_g = S_c^{(m)}$ be the greedy baseline, and let Stage II terminate at a 1-swap stable set $\widehat{S}_c$:

$$F\big((\widehat{S}_c \setminus \{i\}) \cup \{j\}\big) \ \leq \ F(\widehat{S}_c), \qquad \forall i \in \widehat{S}_c, \ \forall j \in \mathcal{T}_c \setminus \widehat{S}_c. \tag{38}$$

Define $A = S_c^{\star} \setminus \widehat{S}_c$ and $B = \widehat{S}_c \setminus S_c^{\star}$ with $|A| = |B| = r$, and fix any bijection $\phi : A \to B$. Let the *local* weak submodularity ratio at base $\widehat{S}_c$ and block $A$ be denoted $\gamma_{\mathrm{wm}}(\widehat{S}_c, A)$; clearly $\gamma_{\mathrm{wm}} \leq \gamma_{\mathrm{wm}}(\widehat{S}_c, A)$ where $\gamma_{\mathrm{wm}}$ is the global infimum (28).

**Lemma C.1** (Add vs. remove under 1-swap stability with approximate DR). *If $F$ satisfies* (37), *then for every $x \in A$ and $y = \phi(x)$,*

$$F(\widehat{S}_c \cup \{x\}) - F(\widehat{S}_c) \ \leq \ \frac{1}{\alpha_{\mathrm{DR}}} \left(F(\widehat{S}_c) - F(\widehat{S}_c \setminus \{y\})\right). \tag{39}$$

*Proof.* Apply (37) with $U = \widehat{S}_c, T = \widehat{S}_c \setminus \{y\}$ and $x \notin \widehat{S}_c$:

$$F((\widehat{S}_c \setminus \{y\}) \cup \{x\}) - F(\widehat{S}_c \setminus \{y\}) \ \geq \ \alpha_{\mathrm{DR}}\big(F(\widehat{S}_c \cup \{x\}) - F(\widehat{S}_c)\big). \tag{40}$$

By 1-swap stability (38), $F((\widehat{S}_c \setminus \{y\}) \cup \{x\}) \leq F(\widehat{S}_c)$, hence $F(\widehat{S}_c) - F(\widehat{S}_c \setminus \{y\}) \ \geq \ \alpha_{\mathrm{DR}}\big(F(\widehat{S}_c \cup \{x\}) - F(\widehat{S}_c)\big)$, which rearranges to (39). $\square$

**Lemma C.2** (Weak-submodularity lower bound).

$$\sum_{x \in A} \big(F(\widehat{S}_c \cup \{x\}) - F(\widehat{S}_c)\big) \ \geq \ \gamma_{\mathrm{wm}}(\widehat{S}_c, A) \big(F(S_c^{\star}) - F(\widehat{S}_c)\big). \tag{41}$$

*Proof.* By the definition of the local ratio at base $\widehat{S}_c$ and block $A$,

$$\sum_{x \in A} \big(F(\widehat{S}_c \cup \{x\}) - F(\widehat{S}_c)\big) \geq \gamma_{\mathrm{wm}}(\widehat{S}_c, A)\big(F(\widehat{S}_c \cup A) - F(\widehat{S}_c)\big). \tag{42}$$

Since $\widehat{S}_c \cup A = S_c^\star \cup B$ and $|B| = r$, by RM in (35) we have $F(S_c^\star \cup B) \geq F(S_c^\star)$, i.e., $F(\widehat{S}_c \cup A) \geq F(S_c^\star)$. Therefore,

$$F(\widehat{S}_c \cup A) - F(\widehat{S}_c) \;\geq\; F(S_c^\star) - F(\widehat{S}_c), \tag{43}$$

which together with the local weak submodularity ratio definition completes the proof. $\qquad\square$

**Lemma C.3** (Curvature upper bound on singleton removals). *If $F$ satisfies (36), then for $U = \widehat{S}_c$ and $V = B$,*

$$\sum_{y \in B} \big(F(\widehat{S}_c) - F(\widehat{S}_c \setminus \{y\})\big) \;\leq\; \frac{1}{1 - \kappa_b}\Big(F(\widehat{S}_c) - F(\widehat{S}_c \setminus B)\Big). \tag{44}$$

**Theorem C.4** (Curvature-augmented Stage II under approximate DR (anchored form)). *Suppose $F$ satisfies RM (35), admits curvature (36) with $\kappa_b < 1$, and satisfies (37) with $\alpha_{\mathrm{DR}} > 0$. Let $\gamma_{\mathrm{wm}} \in (0, 1]$ denote the global weak submodularity ratio (28). Assume additionally that for $B = \widehat{S}_c \setminus S_c^\star$ we have*

$$F(\widehat{S}_c \setminus B) \;\geq\; F(S_g). \tag{45}$$

*Then any 1-swap stable $m$-set $\widehat{S}_c$ returned by Stage II obeys*

$$F(\widehat{S}_c) \;\geq\; F(S_g) \;+\; \theta_\alpha \cdot \big(F(S_c^\star) - F(S_g)\big), \qquad \theta_\alpha \;:=\; \frac{(1 - \kappa_b)\,\alpha_{\mathrm{DR}}\,\gamma_{\mathrm{wm}}}{1 + (1 - \kappa_b)\,\alpha_{\mathrm{DR}}\,\gamma_{\mathrm{wm}}}\,. \tag{46}$$

*Proof.* Summing (39) over $x \in A$ (paired by $y = \phi(x)$) gives

$$\sum_{x \in A} \big(F(\widehat{S}_c \cup \{x\}) - F(\widehat{S}_c)\big) \;\leq\; \frac{1}{\alpha_{\mathrm{DR}}} \sum_{y \in B} \big(F(\widehat{S}_c) - F(\widehat{S}_c \setminus \{y\})\big). \tag{47}$$

Lower-bound the LHS by Lemma C.2 with $\gamma_{\mathrm{wm}}(\widehat{S}_c, A) \geq \gamma_{\mathrm{wm}}$, and upper-bound the RHS by Lemma C.3:

$$\gamma_{\mathrm{wm}}\big(F(S_c^\star) - F(\widehat{S}_c)\big) \;\leq\; \frac{1}{\alpha_{\mathrm{DR}}(1 - \kappa_b)}\Big(F(\widehat{S}_c) - F(\widehat{S}_c \setminus B)\Big). \tag{48}$$

Anchor at the greedy baseline via $\widehat{F}(S) := F(S) - F(S_g)$ (all previous steps are difference-based and thus translation-invariant). Using the additional anchored dominance condition (45), we have

$$F(\widehat{S}_c) - F(\widehat{S}_c \setminus B) \;\leq\; F(\widehat{S}_c) - F(S_g) \;=\; \widehat{F}(\widehat{S}_c). \tag{49}$$

Hence

$$\gamma_{\mathrm{wm}}\big(\widehat{F}(S_c^\star) - \widehat{F}(\widehat{S}_c)\big) \;\leq\; \frac{1}{\alpha_{\mathrm{DR}}(1 - \kappa_b)}\,\widehat{F}(\widehat{S}_c). \tag{50}$$

Rearranging gives $\widehat{F}(\widehat{S}_c) \;\geq\; \dfrac{(1 - \kappa_b)\alpha_{\mathrm{DR}}\gamma_{\mathrm{wm}}}{1 + (1 - \kappa_b)\alpha_{\mathrm{DR}}\gamma_{\mathrm{wm}}}\,\widehat{F}(S_c^\star)$, which is (46). $\qquad\square$

**Remark (anchored dominance).** Condition (45) is a technical *anchored* requirement used only to relate $F(\widehat{S}_c) - F(\widehat{S}_c \setminus B)$ to the baseline gap $F(\widehat{S}_c) - F(S_g)$. Although it depends on the unknown optimal set $S_c^\star$, it serves as a sufficient condition for the anchored guarantee and is empirically observed to hold in our experiments.

**Corollary C.5** (Anchored loss form of Theorem C.4). *With the same assumptions as Theorem C.4 and $F = -\mathcal{L}$, the terminal 1-swap stable set $\widehat{S}_c$ satisfies*

$$\boxed{\;\mathcal{L}(\widehat{S}_c, \mathcal{T}_c) \;\leq\; \mathcal{L}(S_g, \mathcal{T}_c) \;-\; \theta_\alpha\Big(\mathcal{L}(S_g, \mathcal{T}_c) - \mathcal{L}(S_c^\star, \mathcal{T}_c)\Big), \qquad \theta_\alpha = \frac{(1 - \kappa_b)\alpha_{\mathrm{DR}}\gamma_{\mathrm{wm}}}{1 + (1 - \kappa_b)\alpha_{\mathrm{DR}}\gamma_{\mathrm{wm}}}.\;} \tag{51}$$

*In words, Stage II closes a constant fraction $\theta_\alpha$ of the greedy–optimal loss gap.*

**Corollary C.6** (Exact DR (submodular) case). *When $\alpha_{\mathrm{DR}} = \gamma_{\mathrm{wm}} = 1$, (46) simplifies to*

$$F(\widehat{S}_c) \;\geq\; F(S_g) \;+\; \frac{1 - \kappa_b}{2 - \kappa_b}\big(F(S_c^\star) - F(S_g)\big). \tag{52}$$

**Normalized global form.** For a global (nonnegative) normalization, define

$$F_{\min} := \min_{S \subseteq \mathcal{T}_c} F(S), \qquad \widetilde{F}(S) := F(S) - F_{\min} \geq 0. \tag{53}$$

Since all inequalities above depend only on differences of $F$, replacing $F$ by $\widetilde{F}$ leaves $\gamma_{\mathrm{wm}}, \kappa_b, \alpha_{\mathrm{DR}}$ unchanged. Moreover, $\widetilde{F}(\widehat{S}_c) - \widetilde{F}(\widehat{S}_c \setminus B) \leq \widetilde{F}(\widehat{S}_c)$ holds trivially because $\widetilde{F}(\widehat{S}_c \setminus B) \geq 0$. Thus the same derivation yields

$$\widetilde{F}(\widehat{S}_c) \geq \frac{(1 - \kappa_b)\,\alpha_{\mathrm{DR}}\,\gamma_{\mathrm{wm}}}{1 + (1 - \kappa_b)\,\alpha_{\mathrm{DR}}\,\gamma_{\mathrm{wm}}}\,\widetilde{F}(S_c^\star). \tag{54}$$

We emphasize that the anchored form (46) is translation-invariant and preferred in practice.

**Discussion.** (i) $\kappa_b$ quantifies the overlap between singleton removals and their joint removal and remains well-defined even when $F(\varnothing)$ is undefined. (ii) At $\kappa_b = 0$ (no over-counting; e.g., modular objectives), Theorem C.4 yields the classical $1/2$ factor when $\gamma_{\mathrm{wm}} = \alpha_{\mathrm{DR}} = 1$.

**Summary.** Stage I contracts the residual optimality gap at a rate $\left(1 - \frac{\gamma_{\mathrm{wm}}}{m}\right)$ per step, yielding an $e^{-\gamma_{\mathrm{wm}}}$-type reduction after $m$ selections. Stage II performs 1-swap refinement. Under the stated RM/curvature/approximate-DR assumptions (and (45) for the anchored form), the terminal set $\widehat{S}_c$ closes a constant fraction $\theta_\alpha$ of the greedy–optimal utility gap (Eq. (51)).

## D. Symbol Description

To enhance clarity, a detailed description of mathematic symbols in the present study is provided in Table 14.

## E. Implementation Details

**Image Selection Settings.** We use InceptionV3 (Szegedy et al., 2016) as both the feature encoder and classifier (as discussed in G.3, the framework is encoder- and classifier-agnostic). The hyperparameters are set as $\kappa = 1.05$, $\gamma = 0.05$, $T = 20$, $\varepsilon = 10$, $\alpha = 5$, and $\beta = 1000$. The value of $\beta$ depends on the encoder type: 1000 for InceptionV3, 100 for ResNet-50 (He et al., 2016), and 10 for ResNet-34 and GoogLeNet (Szegedy et al., 2015). Following D$^2$C (Huang et al., 2026), each subset is constructed in a class-wise manner, selecting 10, 50, and 100 samples per class, respectively. The input resolutions follow the standard ImageNet settings, with 299×299 for InceptionV3 (Szegedy et al., 2016) and 224×224 for ResNet-34, ResNet-50 (He et al., 2016), and GoogLeNet (Szegedy et al., 2015). These resolutions are used solely for feature extraction during subset selection and are independent of the resolution used in diffusion model training.

**Training Settings.** Following D$^2$C (Huang et al., 2026), we use the Adam optimizer with a fixed learning rate of $1 \times 10^{-4}$ and $(\beta_1, \beta_2) = (0.9, 0.999)$, without applying weight decay. Mixed-precision (fp16) training with gradient clipping is employed for stability and efficiency. Latent representations are pre-computed using the Stable Diffusion VAE (Kingma & Welling, 2013) and decoded via its native decoder. The batch size is set to 128 for all experiments. For a fair comparison, we also adopt the Attach phase from D$^2$C, which enriches each selected image with semantic embeddings extracted from a T5 text encoder (Ni et al., 2022) and visual features obtained from a DINOv2 vision encoder (Oquab et al., 2023). These dual conditional signals provide class-level semantics and instance-level visual priors, thereby enhancing the efficiency and generation quality of diffusion model training.

## F. Sensitivity Analysis of Additional Hyperparameters

We further investigate the sensitivity of the proposed framework to auxiliary hyperparameters. As shown in Table 15, varying the loss weights $\alpha$ and $\beta$ indicates that the method remains stable across a broad range of values. The statistical regularization term $\mathcal{L}_{\mathrm{sta}}$ maintains global feature statistics and stabilizes the overall data distribution. When its weight is too small, the condensed samples exhibit weaker distributional coherence and degraded generative fidelity. Conversely, an excessively large $\alpha$ overemphasizes global alignment, suppressing the OT-driven geometric matching and leading to slight loss of local structural detail. Similarly, the confidence regularization $\mathcal{L}_{\mathrm{conf}}$ enhances semantic consistency by discouraging ambiguous or low-confidence samples. Reducing its weight weakens this semantic constraint, while overly increasing it may bias the subset toward high-confidence regions, leading to a slight reduction in diversity.

*Table 14.* Descriptions of symbols used in the main text.

| Symbol | Definition |
|---|---|
| $\mathcal{T}$ | Full real dataset |
| $\mathcal{S}$ | Condensed (selected) subset |
| $\mathcal{T}_c$ | Real subset for class $c$ |
| $\mathcal{S}_c$ | Condensed subset for class $c$ |
| $\mathcal{T}_{c(e)}$ | Feature embeddings of $\mathcal{T}_c$ |
| $\mathcal{S}_{c(e)}$ | Feature embeddings of $\mathcal{S}_c$ |
| $\mathbf{x}_i$ | Embedding of condensed sample $i$ |
| $\mathbf{y}_j$ | Embedding of real sample $j$ |
| $m$ | Number of samples in $\mathcal{S}_c$ |
| $n$ | Number of samples in $\mathcal{T}_c$ |
| $\mathbf{C}$ | Cost matrix |
| $\mathbf{C}_{ij}$ | Pairwise cost between $\mathbf{x}_i$ and $\mathbf{y}_j$ |
| $\boldsymbol{\pi}$ | Balanced OT transport plan |
| $\mathbf{C}_{\text{aug}}$ | Augmented Cost matrix |
| $\boldsymbol{\Pi}$ | Augmented transport plan with dummy source |
| $\boldsymbol{\mu}$ | Source marginal distribution |
| $\boldsymbol{\nu}$ | Target marginal distribution |
| $\bar{\boldsymbol{\nu}}$ | Reference target marginal |
| $\kappa$ | Capacity scaling factor for one-sided partial OT |
| $\delta$ | Dummy-source cost |
| $\gamma$ | Scaling coefficient for $\delta$ |
| $\varepsilon$ | Entropic regularization weight |
| $\mathbf{K}$ | Gibbs kernel in Sinkhorn iterations |
| $\mathbf{u}$ | Sinkhorn scaling vector on source side |
| $\mathbf{v}$ | Sinkhorn scaling vector on target side |
| $\mathbf{T}_{\text{real}}$ | Real transport matrix after removing dummy mass |
| $\mathcal{L}_{\text{OT}}$ | One-sided partial OT loss |
| $\mathcal{L}_{\text{sta}}$ | Mean–variance regularization loss |
| $\mathcal{L}_{\text{conf}}$ | Confidence regularization loss |
| $\mathcal{L}$ | Overall objective function |
| $\alpha$ | Weight for $\mathcal{L}_{\text{sta}}$ |
| $\beta$ | Weight for $\mathcal{L}_{\text{conf}}$ |
| $\boldsymbol{\mu}_S$ | Mean of features in $\mathcal{S}_c$ |
| $\boldsymbol{\sigma}_S$ | Variance of features in $\mathcal{S}_c$ |
| $\boldsymbol{\mu}_T$ | Mean of features in $\mathcal{T}_c$ |
| $\boldsymbol{\sigma}_T$ | Variance of features in $\mathcal{T}_c$ |
| $p(c|\mathbf{x}_i)$ | Predicted probability of class $c$ for $\mathbf{x}_i$ |
| $x_k$ | Candidate image |
| $\Delta\mathcal{L}(x_k)$ | Marginal gain when adding candidate $\mathbf{x}_k$ |
| $x_t^{\star}$ | Greedy-selected sample at iteration $t$ |
| $\mathcal{S}_c'$ | Temporary subset after swapping one element |
| $\Delta_{i \to j}$ | Objective change for swapping $x_i$ with $x_j$ |
| $s$ | Dummy mass absorbing unmatched targets |
| $r$ | Sinkhorn iteration index |
| $T$ | Total number of Sinkhorn iterations |
| $\mathbf{1}_n$ | All-one vector of length $n$ |
| $\text{Diag}(\cdot)$ | Diagonal matrix operator |

*Table 15.* Impact of loss weights $\alpha$ and $\beta$ on ImageNet (Russakovsky et al., 2015) 256×256 (10K data budget, 100K iterations).

| $\alpha$ | 3 | 5 | 6 | 8 | 10 |
|---|---|---|---|---|---|
| FID | 3.47 | 3.43 | 3.45 | 3.43 | 3.48 |
| IS | 418.1 | 414.3 | 413.7 | 411.4 | 408.6 |
| Precision | 0.80 | 0.78 | 0.78 | 0.77 | 0.76 |
| Recall | 0.27 | 0.28 | 0.28 | 0.28 | 0.29 |

| $\beta$ | 100 | 500 | 1000 | 2000 | 5000 |
|---|---|---|---|---|---|
| FID | 3.50 | 3.45 | 3.43 | 3.48 | 3.56 |
| IS | 391.2 | 405.5 | 414.3 | 421.5 | 427.2 |
| Precision | 0.80 | 0.78 | 0.78 | 0.77 | 0.75 |
| Recall | 0.27 | 0.28 | 0.28 | 0.28 | 0.29 |

*Table 16.* Impact of $\varepsilon$ (entropic regularization) and $T$ (number of Sinkhorn iterations) on ImageNet (Russakovsky et al., 2015) 256×256 (10K data budget, 100K iterations).

| $\varepsilon$ | 5 | 8 | 10 | 12 | 15 |
|---|---|---|---|---|---|
| FID-50K | 3.53 | 3.45 | 3.43 | 3.45 | 3.48 |

| $T$ | 5 | 10 | 20 | 30 | 50 |
|---|---|---|---|---|---|
| FID-50K | 3.55 | 3.47 | 3.43 | 3.42 | 3.43 |

We also analyze the effect of the entropic regularization coefficient $\varepsilon$ and the number of Sinkhorn iterations $T$, as summarized in Table 16. The results demonstrate that the proposed partial OT solver is numerically stable and largely insensitive to these parameters. Smaller regularization or too few iterations may lead to unstable or under-smoothed updates, slightly perturbing the transport plan, whereas larger values produce smoother and more consistent convergence. However, excessively large $T$ yields diminishing returns, providing no further improvement in alignment or fidelity. These observations collectively confirm that our framework exhibits strong robustness to auxiliary hyperparameters, ensuring stable optimization and consistent generative performance across a wide range of configurations.

## G. Further Experimental Analyses

### G.1. Additional 512×512 Results

Beyond the 512×512 DiT-L/2 results at 100K iterations reported in the main text, we further include *SiT-L/2 at 100K* to complement the high-resolution study, as summarized in Table 17. All models are trained from scratch on ImageNet (Russakovsky et al., 2015) using a 10K data budget and evaluated with 10K generated samples. Baseline numbers are taken from D²C (Huang et al., 2026), and both diffusion variants (DiT-L/2 (Peebles & Xie, 2023), SiT-L/2 (Ma et al., 2024)) follow the same protocol as in the main text. Under this additional 512×512 setting, our geometry-aware selection consistently surpasses random sampling and D²C across FID, IS, precision, and recall, confirming the robustness of the proposed selection strategy across different diffusion architectures at high resolution.

### G.2. Comparison with More Baselines

We further compare our geometry-aware dataset condensation with a broader set of representative baselines for diffusion model training. We begin by categorizing existing synthetic data generation approaches into two major families. (i) *Small-scale synthetic-set methods*, which include distribution-matching approaches (Zhao & Bilen, 2023; Wang et al., 2025b; Li et al., 2025b; Cui et al., 2025b; Li et al., 2025c), gradient-matching approaches (Zhao et al., 2020; Kim et al., 2022; Liu et al., 2023a;b), and trajectory-matching approaches (Cazenavette et al., 2022; Zhong et al., 2025; Wang et al., 2025a;

*Table 17.* Comparison of dataset condensation methods on ImageNet (Russakovsky et al., 2015) 512×512 with DiT-L/2 and SiT-L/2 trained for 100K iterations using a 10K data budget. Baseline results are from D²C (Huang et al., 2026).

| Model | Method | Data Budget | Iteration | FID↓ | IS↑ | Precision↑ | Recall↑ |
|---|---|---|---|---|---|---|---|
| DiT-L/2 | Random | 0.8% (10K) | 100K | 24.8 | 74.3 | 0.65 | 0.42 |
| DiT-L/2 | D²C (Huang et al., 2026) | 0.8% (10K) | 100K | 14.8 | 109.2 | 0.63 | 0.52 |
| DiT-L/2 | Ours | 0.8% (10K) | 100K | **6.2** | **451.0** | **0.81** | **0.67** |
| SiT-L/2 | Random | 0.8% (10K) | 100K | 13.3 | 197.1 | 0.69 | 0.68 |
| SiT-L/2 | D²C (Huang et al., 2026) | 0.8% (10K) | 100K | 9.1 | 261.7 | 0.72 | 0.34 |
| SiT-L/2 | Ours | 0.8% (10K) | 100K | **5.8** | **461.6** | **0.81** | **0.66** |

*Table 18.* Comparison of different dataset condensation methods using DiT-L/2 (Peebles & Xie, 2023) on ImageNet (Russakovsky et al., 2015) at 256×256 resolution. All methods are trained for 100K iterations with a 0.8% (10K) data budget and evaluated under two settings using 10K and 50K generated samples for FID and IS computation. For fairness, all baseline methods adopt the same attach phase as $D^2C$ (Huang et al., 2026). Best results are shown in **bold**.

| Method | Data Budget | Eval. Samples | FID↓ | IS↑ | Precision↑ | Recall↑ |
|---|---|---|---|---|---|---|
| CCS (Zheng et al., 2023) | 0.8% (10K) | 10K | 7.81 | 373.3 | 0.77 | 0.58 |
| EDC (Shao et al., 2024) | 0.8% (10K) | 10K | 31.7 | 161.5 | 0.52 | 0.63 |
| RDED (Sun et al., 2024a) | 0.8% (10K) | 10K | 24.1 | **491.0** | 0.80 | 0.25 |
| IGD (Chen et al., 2025) | 0.8% (10K) | 10K | 15.92 | 433.9 | **0.83** | 0.41 |
| Ours | 0.8% (10K) | 10K | **5.76** | 430.2 | 0.78 | **0.69** |
| CCS (Zheng et al., 2023) | 0.8% (10K) | 50K | 5.45 | 364.9 | 0.77 | 0.21 |
| EDC (Shao et al., 2024) | 0.8% (10K) | 50K | 29.4 | 161.7 | 0.52 | 0.20 |
| RDED (Sun et al., 2024a) | 0.8% (10K) | 50K | 20.90 | **482.7** | 0.82 | 0.05 |
| IGD (Chen et al., 2025) | 0.8% (10K) | 50K | 12.67 | 412.5 | **0.85** | 0.10 |
| Ours | 0.8% (10K) | 50K | **3.43** | 414.3 | 0.78 | **0.28** |

*Table 19.* Comparison of different dataset condensation methods using SiT-L/2 (Ma et al., 2024) on ImageNet (Russakovsky et al., 2015) at 256×256 resolution. All methods are trained for 100K iterations with a 0.8% (10K) data budget and evaluated under two settings using 10K and 50K generated samples for FID and IS computation. For fairness, all baseline methods adopt the same attach phase as $D^2C$ (Huang et al., 2026). Best results are shown in **bold**.

| Method | Data Budget | Eval. Samples | FID↓ | IS↑ | Precision↑ | Recall↑ |
|---|---|---|---|---|---|---|
| CCS (Zheng et al., 2023) | 0.8% (10K) | 10K | 9.52 | 377.0 | 0.78 | 0.58 |
| EDC (Shao et al., 2024) | 0.8% (10K) | 10K | 29.3 | 171.8 | 0.55 | 0.64 |
| RDED (Sun et al., 2024a) | 0.8% (10K) | 10K | 26.0 | **490.3** | 0.82 | 0.26 |
| IGD (Chen et al., 2025) | 0.8% (10K) | 10K | 16.1 | 427.5 | **0.86** | 0.43 |
| Ours | 0.8% (10K) | 10K | **5.91** | 423.2 | 0.78 | **0.70** |
| CCS (Zheng et al., 2023) | 0.8% (10K) | 50K | 6.54 | 373.1 | 0.78 | 0.19 |
| EDC (Shao et al., 2024) | 0.8% (10K) | 50K | 26.7 | 169.9 | 0.54 | 0.22 |
| RDED (Sun et al., 2024a) | 0.8% (10K) | 50K | 22.7 | **483.1** | 0.82 | 0.06 |
| IGD (Chen et al., 2025) | 0.8% (10K) | 50K | 13.2 | 427.0 | **0.86** | 0.12 |
| Ours | 0.8% (10K) | 50K | **3.56** | 415.4 | 0.78 | **0.27** |

Zhao et al., 2025). These methods typically operate at low resolution (e.g., CIFAR-level) due to their substantial memory and computational overhead. (ii) *Large-scale synthetic-set methods* designed for ImageNet-level settings, consisting of uni-level or model-inversion-based approaches (Yin et al., 2023; Cui et al., 2025a; Sun et al., 2024a; Shao et al., 2024) and generative-model-based approaches (Gu et al., 2024; Abbasi et al., 2024; Su et al., 2024; Chen et al., 2025). Most real subset selection baselines with publicly available implementations have already been compared in the main text. As a complement, we additionally consider selection with additional image-level processing (RDED (Conforti & Cornuéjols, 1984)) and traditional coreset selection methods equipped with the attachment phase from $D^2C$ (Huang et al., 2026).

**Why small-scale synthetic set generation methods are omitted here.** Classical distribution-matching, gradient-matching, and trajectory-matching approaches construct a synthetic set by iteratively refining pixel values of a small set of images. They typically treat synthetic samples as fixed entities that must be stored and optimized throughout training, require exhaustive pixel-level updates coupled with backpropagation through the training trajectory, and repeatedly access the full real dataset for each refinement step. These three factors lead to substantial time and memory overhead, which in practice confines such methods to small-scale benchmarks (e.g., CIFAR, Tiny-ImageNet) at low resolution. Directly scaling these synthetic-set generation pipelines to ImageNet-1K at 256×256 or 512×512 would require prohibitive GPU memory and wall-clock time, and existing works do not report competitive performance in such regimes. For this reason, we do not include these small-scale synthetic baselines in our large-scale evaluation and instead focus on methods that have been explicitly designed or adapted for ImageNet-level settings. However, some optimization objectives from distribution-matching-based methods are amenable to integration into our alignment-driven selection framework, and the corresponding results are presented later in Appendix G.5.

**More Real subset selection baselines.** Tables 18 and 19 summarize results on ImageNet-1K at 256×256 using DiT-L/2 (Peebles & Xie, 2023) and SiT-L/2 (Ma et al., 2024), respectively, under a 0.8% (10K) data budget and 100K training iterations.

*Table 20.* Evaluation on ImageNet (Russakovsky et al., 2015) at $256 \times 256$ using DiT-L/2 (Peebles & Xie, 2023). K-Centers[†] and K-Medoids[†] denote classical coreset selection methods conducted in the feature space and augmented with the same attachment phase as $D^2C$ (Huang et al., 2026) to enable fair comparison under diffusion model training. Best results are shown in **bold**.

| Method | Data Budget | Eval. Samples | FID↓ | IS↑ | Precision↑ | Recall↑ |
|---|---|---|---|---|---|---|
| K-Centers[†] (Gonzalez, 1985) | 0.8% (10K) | 50K | 39.67 | 142.1 | 0.27 | **0.34** |
| K-Mediods[†] (Kaufman & Rousseeuw, 1987) | 0.8% (10K) | 50K | 4.13 | 315.7 | 0.77 | 0.24 |
| Ours | 0.8% (10K) | 50K | **3.43** | **414.3** | **0.78** | 0.28 |

For our task, RDED (Sun et al., 2024a) exhibits a similar behavior in that it performs sample selection solely based on confidence scores. This strategy yields a very high Inception Score, but the absence of any distributional alignment leads to poor FID, and the lack of geometric constraints results in inadequate coverage, reflected by its very low Recall.

For dataset quantization methods, we compare DQ (Zhou et al., 2023) in the main text. As DQAS (Zhao et al., 2024) and ADQ (Li et al., 2025a) do not release their code publicly, we are unable to conduct a direct comparison with them.

As a complementary direction, we also consider real subset selection methods originating from data pruning. We include CCS (Zheng et al., 2023) as a representative pruning-based method. More recent pruning approaches such as TDDS (Zhang et al., 2024c) could also be relevant, but their ImageNet-1K implementations are not publicly available, preventing a fair comparison in our setting. CCS evaluates per-sample importance based on classification-oriented criteria such as correctness, forgetting events, prediction margin, and EL2N scores, aiming to retain examples that are most influential for supervised discriminative learning. However, these indicators are tailored to classification dynamics and do not capture the geometric, statistical, or manifold-level properties required for diffusion training; consequently, CCS-selected subsets fail to preserve distributional structure, exhibit limited coverage, and yield suboptimal FID, IS, precision, and recall compared to our geometry-aware formulation.

Apart from comparisons with direct coreset selection methods in Figures 1 and 5, we further compare against coreset selection methods equipped with the attachment phase from $D^2C$ (Huang et al., 2026) and include stronger baselines in Table 20. The K-center method (Gonzalez, 1985; Sener & Savarese, 2018), with its farthest-point greedy criterion, tends to push selected samples toward the boundary of the embedding space, thereby overemphasizing extreme and low-density regions. While this strategy improves geometric coverage, it often sacrifices representativeness with respect to the underlying data distribution. In contrast, K-Medoids (Kaufman & Rousseeuw, 1987) selects representative samples by minimizing the average distance between data points and their assigned medoids. Although this objective favors central and high-density regions, it can lead to insufficient coverage of the data manifold, particularly in the presence of multi-modal distributions. As a result, K-Medoids may overlook structurally important but less populated regions, yielding subsets that are biased toward the most dominant mode.

**Large-scale synthetic set generation baselines.** Large-scale synthetic set generation methods can be divided into two types: model-inversion-based approaches and generative-model-based methods. Model-inversion-based approaches compress the real dataset into a compact model representation, eliminating the need for real data during image refinement. We include a representative baseline EDC (Shao et al., 2024) for comparison.

Generative-model-based approaches leverage pretrained generative models to avoid pixel-level refinements. In this category, we include IGD (Chen et al., 2025) as a recent representative method.

EDC (Shao et al., 2024) extends RDED by performing pixel-level optimization to align the condensed set with the stored BatchNorm statistics of the real dataset. However, this continuous pixel-wise refinement disrupts the visual characteristics and structural integrity of the images, leading to significantly degraded FID and Inception Score. Moreover, because the optimization is unconstrained with respect to the real data manifold, it produces samples that deviate from plausible image space, resulting in poor Precision.

Because our method jointly incorporates geometric distribution information, statistical alignment, and semantic confidence cues, and performs discrete optimization strictly over real images on the true data manifold, it avoids these issues and achieves a well-balanced trade-off across fidelity, coverage, and semantic quality.

Generative-model-based condensation inherently suffers from two limitations. Training or adapting large generators to new domains is computationally expensive and often yields domain-specific priors that generalize poorly beyond the source distribution. Moreover, generative-model-based condensation exhibits a characteristic metric profile because the synthesized

*Table 21.* Ablation on the feature *encoder / classifier* used for selection on ImageNet (Russakovsky et al., 2015) at 256×256. All rows train the same diffusion model under a 10K (0.8%) budget for 100K iterations, and are evaluated with 50K generated samples.

| Method | Network | Data Budget | FID↓ | IS↑ | Precision↑ | Recall↑ |
|---|---|---|---|---|---|---|
| D²C (Huang et al., 2026) | DiT-XL/2 (Peebles & Xie, 2023) | 0.8% (10K) | 4.20 | 283.6 | 0.72 | 0.24 |
| Ours | ResNet-34 (He et al., 2016) | 0.8% (10K) | 3.42 | 385.4 | 0.78 | 0.27 |
| | ResNet-50 (He et al., 2016) | 0.8% (10K) | 3.47 | 407.1 | 0.77 | 0.29 |
| | GoogLeNet (Szegedy et al., 2015) | 0.8% (10K) | 3.44 | 396.8 | 0.78 | 0.28 |
| | InceptionV3 (Szegedy et al., 2015) | 0.8% (10K) | 3.43 | 414.3 | 0.78 | 0.28 |

set is drawn from the generator's learned manifold rather than the true data manifold. Without an explicit constraint to align these manifolds, a domain gap emerges, which inflates FID. At the same time, sampling concentrates probability mass on high-density modes that the generator models confidently. This mode concentration yields high Precision and a strong Inception Score, yet simultaneously induces mode collapse, omitting low-density and boundary regions of the real distribution. The resulting lack of coverage depresses Recall and further contributes to a higher FID through diversity loss and second-order statistic mismatch.

In short, local fidelity around a few dominant modes (high Precision/IS) coexists with global misalignment and under-coverage (low Recall, elevated FID) due to the uncorrected domain gap between the generator's manifold and the real manifold. In contrast, our geometry-aware condensation directly selects informative real samples without relying on generator synthesis or retraining. This design preserves both visual fidelity and manifold geometry of the original dataset, ensuring balanced coverage and accurate statistical representation. Consequently, the condensed subset maintains high precision while significantly improving recall and FID, demonstrating better data efficiency and overall generative performance compared to generative-model-based condensation approaches such as IGD (Chen et al., 2025).

**Summary.** Overall, the comparisons in Tables 18, 19 and 20 highlight that synthetic set generation methods either fail to scale to ImageNet-1K or exhibit undesirable trade-offs between fidelity and coverage. Real subset selection also does not fully address geometric alignment. By directly selecting informative real samples under a geometry-aware objective, our method preserves both visual fidelity and manifold structure of the original dataset, achieving superior FID and recall while maintaining competitive precision, recall and IS across different diffusion backbones and evaluation protocols. These results demonstrate that geometry-aware real subset selection is a more effective and scalable solution for large-scale diffusion training than existing synthetic set generation and real image selection based approaches.

### G.3. Ablation of Feature Encoder and Classifier

We perform alignment in a pretrained representation space that is widely used to preserve semantic neighborhoods and perceptual similarity. Under this metric, preserving neighborhood structure provides a practical surrogate for preserving the support geometry relevant to likelihood-based generative training.

To investigate the impact of the feature encoder and its paired classifier on the selection process, we conduct an ablation study using three representative backbones with different architectures and capacities. Specifically, the default feature extractor is replaced with ResNet-34 (He et al., 2016), ResNet-50 (He et al., 2016), GoogLeNet (Szegedy et al., 2015), and InceptionV3 (Szegedy et al., 2016), while all other settings remain unchanged. As shown in Table 21, all variants achieve comparable results in terms of FID, Inception Score, precision, and recall, exhibiting only negligible variations. This consistency demonstrates that the proposed framework is *encoder-agnostic and classifier-agnostic*, since the condensation performance depends primarily on the geometric and statistical structure of the learned embedding space rather than the specific encoder design. Each feature encoder is paired with its corresponding frozen classifier head, and no additional supervision or fine-tuning is introduced. The classifier serves solely as a semantic probe for computing confidence regularization, ensuring consistent conditions across all configurations. These results confirm that the method generalizes well to different pretrained encoders (classifiers) and maintains stable performance across different backbone designs.

### G.4. Further Analysis of One-sided Partial Optimal Transport

As illustrated in Fig. 5, we further visualize a 2D toy case to compare the geometric behaviors of classical balanced OT and our one-sided partial OT. Under the balanced formulation, the transport plan enforces uniform mass matching across all real samples, including those lying in peripheral low-density regions. Such peripheral samples distort transport distances and

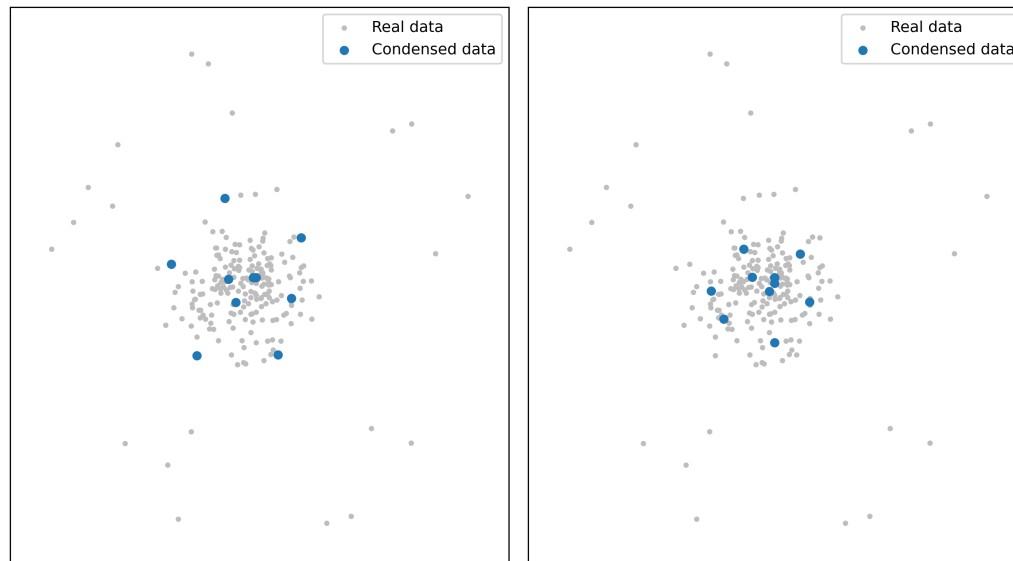

*Figure 5.* Visualization of the condensed subset selected by classical balanced OT (left) and one-sided partial OT (right) on a 2D toy dataset. Grey dots denote real samples, and blue dots denote selected condensed samples.

mislead the alignment away from the high-density core manifold, resulting in redundant assignments and an over-dispersed condensed subset. In contrast, partial OT introduces a capacity relaxation that allows unmatched mass in these low-density areas, effectively suppressing spurious couplings and concentrating the condensed subset on the high-density manifold. This selective relaxation yields a more compact and geometry-consistent representation, highlighting the superiority of one-sided partial transport in mitigating structural imbalance and preserving the intrinsic data geometry.

As reported in Tables 22 and 23, we further quantify the density–mass relationship under one-sided partial OT. Compared to the balanced counterpart, POT consistently allocates lower mass fraction and smaller average cost $C$ in low-density bins (e.g., bins 0–2), while slightly increasing both in high-density regions. This pattern indicates that POT adaptively suppresses unreliable transport in sparse areas, thereby emphasizing dense and structurally coherent regions. Such redistribution aligns with the geometric intuition of partial matching, where the capacity relaxation effectively filters peripheral noise and promotes compact manifold coverage. Overall, this selective mass reallocation demonstrates that one-sided partial OT achieves a better balance between coverage and compactness, faithfully capturing the intrinsic data geometry while avoiding over-assignment to outliers.

### G.5. Adapting Existing Distribution-Matching Objectives within Our Alignment-Driven Framework

Most existing synthetic set generation methods are not directly applicable to diffusion model training, as they rely on continuous sample synthesis and are primarily designed for discriminative or reconstruction-oriented objectives. Such synthesized samples often exhibit limited fidelity and are ill-suited for likelihood-based generative modeling. Nevertheless, a subset of these methods is built upon distribution-matching principles, which are conceptually aligned with the goal of diffusion model training.

In this subsection, we demonstrate that distribution-matching objectives originally proposed for synthetic set generation can be seamlessly incorporated into our alignment-driven discrete subset selection framework. Specifically, we retain the proposed discrete optimization pipeline and selection strategy, while replacing our alignment objective with alternative distribution-matching objectives, including those from DM (Zhao & Bilen, 2023), M3D (Zhang et al., 2024a), and M3D+OPTICAL (Cui et al., 2025b). All other components of the framework remain unchanged. This setting allows us to isolate the effect of the objective itself and evaluate its compatibility with discrete real subset selection.

Table 24 reports the resulting performance on ImageNet. While these adapted objectives yield reasonable results when instantiated within our framework, they consistently underperform our proposed formulation. This suggests that our alignment-driven objective is better suited for discrete subset selection under diffusion model training. At the same time, the competitive performance of these adapted objectives indicates that our framework serves as an effective bridge between

*Table 22.* Density-bin comparison between POT ($\kappa = 1.05$) and classical balanced OT for class 0 ($k_{\text{NN}} = 10$, 20 bins).

| Bin | Density Quantile Range | N | POT ($\kappa = 1.05$) | | OT ($\kappa = 1.0$) | |
|---|---|---|---|---|---|---|
| | | | Mass Fraction | Avg. $C$ | Mass Fraction | Avg. $C$ |
| 0 | [0.0429, 0.0741] | 65 | 0.0068 | 204.1019 | 0.0500 | 303.6486 |
| 1 | [0.0741, 0.0961] | 65 | 0.0456 | 158.8873 | 0.0500 | 163.7466 |
| 2 | [0.0961, 0.1109] | 65 | 0.0526 | 117.6554 | 0.0500 | 117.3249 |
| 3 | [0.1109, 0.1233] | 65 | 0.0526 | 94.1423 | 0.0500 | 93.3965 |
| 4 | [0.1233, 0.1355] | 65 | 0.0526 | 91.5579 | 0.0500 | 90.9135 |
| 5 | [0.1355, 0.1438] | 65 | 0.0527 | 77.3135 | 0.0500 | 76.4463 |
| 6 | [0.1438, 0.1517] | 65 | 0.0527 | 77.7354 | 0.0500 | 76.8634 |
| 7 | [0.1517, 0.1589] | 65 | 0.0527 | 69.8942 | 0.0500 | 68.8171 |
| 8 | [0.1589, 0.1672] | 65 | 0.0527 | 67.3078 | 0.0500 | 66.1017 |
| 9 | [0.1672, 0.1724] | 65 | 0.0527 | 64.1577 | 0.0500 | 62.9709 |
| 10 | [0.1724, 0.1786] | 65 | 0.0527 | 62.1195 | 0.0500 | 60.9175 |
| 11 | [0.1786, 0.1846] | 65 | 0.0527 | 56.5397 | 0.0500 | 55.3196 |
| 12 | [0.1846, 0.1907] | 65 | 0.0527 | 53.5391 | 0.0500 | 52.3477 |
| 13 | [0.1907, 0.1979] | 65 | 0.0527 | 53.2931 | 0.0500 | 52.0506 |
| 14 | [0.1979, 0.2049] | 65 | 0.0527 | 51.6530 | 0.0500 | 50.3091 |
| 15 | [0.2049, 0.2132] | 65 | 0.0527 | 48.0480 | 0.0500 | 46.8166 |
| 16 | [0.2132, 0.2216] | 65 | 0.0527 | 46.8586 | 0.0500 | 45.5787 |
| 17 | [0.2216, 0.2292] | 65 | 0.0527 | 42.4270 | 0.0500 | 41.2335 |
| 18 | [0.2292, 0.2468] | 65 | 0.0527 | 39.1556 | 0.0500 | 37.9536 |
| 19 | [0.2468, 0.2878] | 65 | 0.0527 | 35.5625 | 0.0500 | 34.4713 |

synthetic set generation objectives and real subset selection, providing a unified carrier for evaluating and comparing diffusion-oriented alignment criteria.

We further analyze the behavior of different distribution-matching objectives when instantiated within our framework. DM (Zhao & Bilen, 2023) adopts an MMD-based objective that primarily enforces consistency of first-order statistics. As a result, the selected samples tend to concentrate around the central region of the data distribution, leading to limited coverage and weaker representativeness, which in turn degrades generative performance. M3D (Zhang et al., 2024a) extends this formulation by computing MMD in the kernel Hilbert space, thereby encouraging consistency of higher-order statistics beyond the mean. This modification substantially strengthens distributional alignment and yields clear improvements over DM in terms of both coverage and generation quality. Building upon this, M3D+OPTICAL (Cui et al., 2025b) further incorporates optimal transport to reallocate sample contributions, explicitly adjusting the matching between the selected subset and the full distribution. This transport-based correction enhances set-level alignment and leads to additional performance gains compared to MMD-based objectives alone. Despite these improvements, these objectives do not explicitly enforce semantic consistency of the selected samples. Consequently, while distributional alignment is improved, semantic reliability is not guaranteed, which is reflected in relatively lower inception scores. Moreover, the use of full matching in these objectives can overemphasize peripheral or low-density regions, introducing sensitivity to boundary samples that may partially distort the selected set. In contrast, our method combines geometry-aware alignment with semantic regularization and adopts a one-sided partial matching formulation, allowing it to focus on representative regions while preserving semantic fidelity, thereby achieving consistently superior performance.

### G.6. Clarification for the Meaning of "Geometry"

Our method is not a generic metric matching approach but rather a distributional geometry alignment framework specifically designed for the score-matching objective of diffusion models.

In our paper, we use the term "geometry" not in the strict differential-geometric sense, e.g., curvature or topology, but to refer to the **distributional support geometry** of the data in representation space. This concept encompasses the **local neighborhood structure**, **mode coverage**, and **support allocation**. Specifically, this is the type of geometry emphasized in Section 3.2, which is crucial for diffusion models to learn an accurate score function without distorting the underlying manifold. In this context, preserving relative relationships and local structure takes precedence over scalar ranking, as these aspects are directly relevant to the model's ability to capture the true data distribution.

In modern geometric analysis, this view is consistent with the classical understanding that the $L_2$ Wasserstein distance is not merely a distance penalty, but endows the space of probability measures with a geometric structure (**??**). The cost matrix

*Table 23.* Density-bin comparison between POT ($\kappa = 1.05$) and classical balanced OT for class 1 ($k_{\text{NN}} = 10$, 20 bins).

| Bin | Density Quantile Range | N | POT ($\kappa = 1.05$) | | OT ($\kappa = 1.0$) | |
|---|---|---|---|---|---|---|
| | | | Mass Fraction | Avg. $C$ | Mass Fraction | Avg. $C$ |
| 0 | [0.0492, 0.0654] | 65 | 0.0094 | 269.4661 | 0.0500 | 330.3231 |
| 1 | [0.0654, 0.0763] | 65 | 0.0424 | 234.1719 | 0.0500 | 239.7473 |
| 2 | [0.0763, 0.0869] | 65 | 0.0517 | 190.7487 | 0.0500 | 188.7713 |
| 3 | [0.0869, 0.0979] | 65 | 0.0527 | 152.7371 | 0.0500 | 149.8369 |
| 4 | [0.0979, 0.1066] | 65 | 0.0527 | 135.2751 | 0.0500 | 131.3823 |
| 5 | [0.1066, 0.1139] | 65 | 0.0527 | 119.0013 | 0.0500 | 115.9252 |
| 6 | [0.1139, 0.1209] | 65 | 0.0527 | 105.5147 | 0.0500 | 102.6385 |
| 7 | [0.1209, 0.1281] | 65 | 0.0527 | 96.2394 | 0.0500 | 93.5553 |
| 8 | [0.1281, 0.1350] | 65 | 0.0527 | 90.6207 | 0.0500 | 88.0241 |
| 9 | [0.1350, 0.1435] | 65 | 0.0527 | 83.5750 | 0.0500 | 81.5046 |
| 10 | [0.1435, 0.1505] | 65 | 0.0527 | 71.8745 | 0.0500 | 70.3046 |
| 11 | [0.1505, 0.1575] | 65 | 0.0527 | 72.7010 | 0.0500 | 71.1531 |
| 12 | [0.1575, 0.1651] | 65 | 0.0527 | 68.6823 | 0.0500 | 67.4286 |
| 13 | [0.1651, 0.1706] | 65 | 0.0527 | 65.1971 | 0.0500 | 63.7957 |
| 14 | [0.1706, 0.1766] | 65 | 0.0527 | 61.3405 | 0.0500 | 60.1314 |
| 15 | [0.1766, 0.1826] | 65 | 0.0527 | 59.9761 | 0.0500 | 58.8863 |
| 16 | [0.1826, 0.1900] | 65 | 0.0527 | 58.6127 | 0.0500 | 57.4309 |
| 17 | [0.1900, 0.1987] | 65 | 0.0527 | 55.1397 | 0.0500 | 54.0474 |
| 18 | [0.1987, 0.2096] | 65 | 0.0527 | 50.5011 | 0.0500 | 49.5143 |
| 19 | [0.2096, 0.2382] | 65 | 0.0527 | 43.4175 | 0.0500 | 42.5779 |

*Table 24.* Compatibility of our alignment-driven discrete subset selection framework with existing synthetic set generation objectives, including DM (Zhao & Bilen, 2023), M3D (Zhang et al., 2024a), and M3D+OPTICAL (Cui et al., 2025b). These objectives are instantiated as alignment criteria within our discrete optimization framework and evaluated on ImageNet (Russakovsky et al., 2015) at $256 \times 256$ using DiT-L/2 (Peebles & Xie, 2023).

| Method | Data Budget | Eval. Samples | FID↓ | IS↑ | Precision↑ | Recall↑ |
|---|---|---|---|---|---|---|
| DM (Zhao & Bilen, 2023) | 0.8% (10K) | 50K | 4.34 | 321.9 | **0.78** | 0.23 |
| M3D (Zhang et al., 2024a) | 0.8% (10K) | 50K | 3.83 | 317.1 | 0.75 | 0.24 |
| M3D+OPTICAL (Cui et al., 2025b) | 0.8% (10K) | 50K | 3.72 | 312.6 | 0.76 | 0.26 |
| Ours | 0.8% (10K) | 50K | **3.43** | **414.3** | **0.78** | **0.28** |

defines the distances between samples, but it is the OT/POT mechanism that realizes the alignment of data distributions in a way that captures the distributional geometry in representation space. One-sided POT is particularly important here. Under extreme condensation budgets, balanced OT enforces symmetric full mass matching between real and selected samples and may bias the selected set away from the core manifold. Our one-sided POT relaxes this requirement and suppresses such spurious couplings. By concentrating mass on the high-density, stable support, POT ensures that the geometry-preserving alignment is directly aligned with the score-matching objective of diffusion models, which requires maintaining local structure and relative relationships.

We directly validate this notion of geometry in the paper, rather than merely reporting final generative metrics. Our analysis includes manifold coverage, mean nearest-neighbor distance, and the effect of POT in avoiding over-assignment to peripheral regions, as shown in Figure 5, Figure 7, Table 22, and Table 23. All of these measures are designed to assess the preservation of neighborhood- and support-level geometry in representation space.

Here we report additional metrics that are designed to quantify how well a selected subset preserves the **neighborhood geometry** of the full dataset:

- **Mean nearest selected distance** measures the average distance from each full-dataset sample to its nearest selected sample, with lower values indicating that the selected subset stays closer to the original data manifold.

- **Coverage at $k$-radius** measures the proportion of samples whose nearest selected sample falls within that sample's own local $k$-NN radius, such as the 10-NN radius, so higher values indicate better local coverage of the original distribution.

- **Nearest selected in $k$-NN rate** further examines whether the nearest selected sample truly belongs to the sample's original $k$ nearest neighbors, with higher values reflecting better preservation of authentic local neighbor relationships.

- **Mean rank of nearest selected** computes the average rank position of the nearest selected sample in the full neighbor

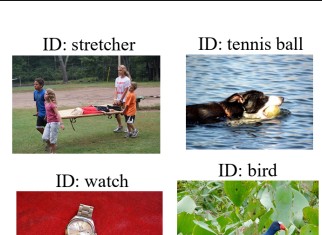
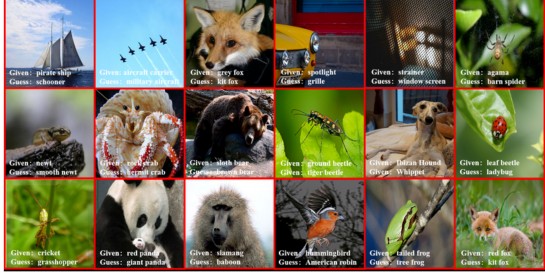
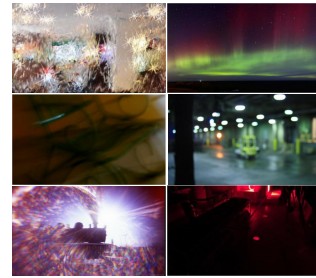

(a) Multiple salient objects      (b) Mislabeled samples      (c) Severely degraded samples

*Figure 6.* Representative failure cases in large-scale datasets such as ImageNet (Russakovsky et al., 2015) that motivate confidence regularization. (a) Images containing multiple salient objects, where the labeled category is not visually dominant (images adapted from (Miyai et al., 2025)). (b) Genuinely mislabeled samples, where the assigned category is inconsistent with the actual visual content (images adapted from (Zheng et al., 2025)). (c) Severely degraded samples with extreme blur, occlusion, or scale, where the labeled object is barely recognizable. While only (b) constitutes true annotation errors, all three cases can introduce unreliable geometric anchors and distort distribution alignment if treated equally.

ordering of each point, where lower values mean that selected samples tend to remain top-ranked local neighbors rather than merely nearby points.

These metrics are particularly suitable for quantifying neighborhood geometry preservation because they evaluate complementary aspects of local structure, including distance proximity, local coverage, neighbor identity consistency, and relative rank ordering. Results in Table 25 show that our method preserves local neighborhood geometry much better than D$^2$C.

*Table 25.* Neighborhood geometry preservation of selected subsets. Lower mean nearest distance and mean nearest selected rank are better, while higher coverage and nearest-selected-in-10NN rate are better.

| Subset | Mean Nearest Distance ↓ | Coverage@10 -radius ↑ | Nearest-Selected -in-10NN ↑ | Mean Nearest Selected Rank ↓ |
|---|---|---|---|---|
| D$^2$C (Class 0) | 8.9 | 0.057 | 0.051 | 125.59 |
| Ours (Class 0) | 6.7 | 0.140 | 0.138 | 103.40 |
| D$^2$C (Class 1) | 9.5 | 0.043 | 0.036 | 204.39 |
| Ours (Class 1) | 7.8 | 0.150 | 0.144 | 103.79 |

## G.7. Why Confidence Regularization Matters: Low-confidence Samples Undermine Geometry-Preserving Alignment

**Motivation.** Our one-sided POT objective is designed to preserve feature-space geometry, yet large-scale datasets such as ImageNet inevitably contain images whose *semantic evidence* for the nominal class label is unreliable. As illustrated in Fig. 6, representative cases include (i) scenes with multiple salient objects where the labeled category is not visually dominant; (ii) genuinely mislabeled images where the assigned category is inconsistent with the actual visual content; and (iii) severely degraded images where extreme blur, occlusion, or scale makes the labeled object barely recognizable. Importantly, these cases are not simply "hard" classification instances; for conditional generative modeling they correspond to unreliable (and sometimes contradictory) conditioning signals, and treating them on par with clean data can bias distribution-preserving condensation.

**A concrete example: how low-confidence selections harm geometric alignment.** Consider a class $c$ (e.g., `golden retriever`) and a candidate image whose visual content is dominated by background clutter or co-occurring objects, or whose labeled object is barely visible. A pretrained classifier typically assigns a low confidence $p(c \mid \cdot)$ to such an image, reflecting weak class-consistent evidence. In the embedding space induced by the pretrained encoder, these images are often mapped to regions closer to background-driven patterns or class-boundary areas than to the core class-conditional manifold. The key issue arises when such a sample is *selected* into the condensed set $S_c$: it becomes a source-side *geometric anchor* whose pairwise distances to other selected samples contribute to the global geometry used for alignment. Because its embedding is semantically unreliable and often lies off the class-conditional manifold, it can distort the overall distance structure of the selected set, shift the effective geometric center, and consume subset capacity to represent background-driven or boundary regions. Under tight condensation budgets, even a small number of such unreliable anchors can substantially

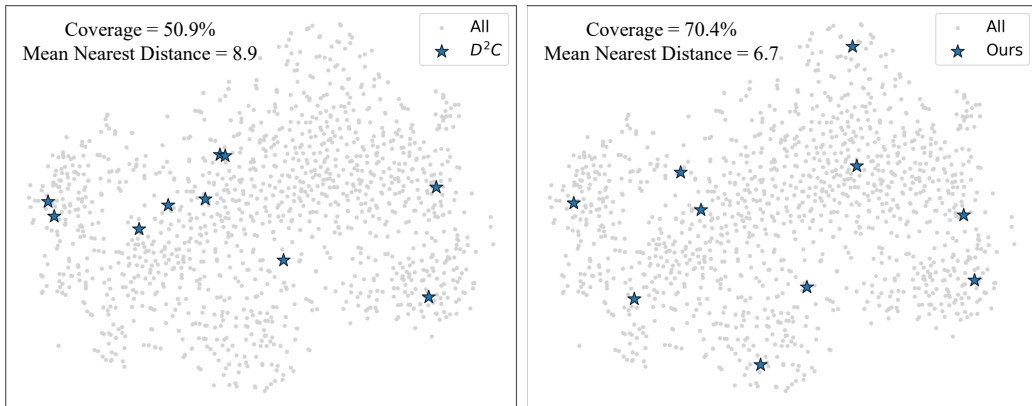

*Figure 7.* t-SNE visualization comparing D$^2$C (Huang et al., 2026) (left) and our method (right). We visualize the real data (gray) from class 0 (tench) of ImageNet and the selected subset samples (blue). In this visualization, D$^2$C selects samples by uniformly spacing along a scalar difficulty ranking, which can lead to uneven coverage in the embedding distribution. In contrast, our one-sided partial OT–guided alignment-driven selection produces subsets that appear more broadly distributed, providing qualitative intuition for the improved alignment achieved by our method.

reduce intra-class coherence and bias the selected set away from the dominant manifold, which in turn compromises distribution-preserving alignment for diffusion training.

**Why this is specifically harmful for diffusion-oriented condensation.** Diffusion training is driven by likelihood maximization and is therefore highly sensitive to biases in the *support* and *global geometry* of the training distribution. Unlike discriminative training, where a small number of atypical hard examples may help sharpen decision boundaries, diffusion models require the training subset to faithfully represent the class-conditional data manifold. When semantically unreliable images enter $S_c$, they effectively behave as geometric outliers: their embeddings distort the global distance structure of the subset and misallocate limited subset capacity toward background-driven or semantically inconsistent regions, rather than the dominant class manifold. This effect is particularly pronounced under tight condensation budgets, where each selected sample carries substantial representational weight. Moreover, severely degraded images provide corrupted or information-poor visual evidence, which is fundamentally misaligned with the objective of learning high-fidelity generative distributions and can directly impair likelihood-based diffusion training. As a result, retaining such samples degrades distributional alignment and manifests as reduced semantic clarity and lower distribution fidelity in generation.

**Confidence regularization as a semantic reliability constraint for discrete selection.** We therefore introduce the confidence regularization

$$\mathcal{L}_{\mathrm{conf}} = \frac{1}{m} \sum_{i=1}^{m} - \log p(c \mid \mathbf{x}_i),$$

which penalizes selecting samples that a pretrained classifier deems weakly supported by the nominal class label. In practice, low classifier confidence correlates well with the failure cases illustrated in Fig. 6, including mislabeled samples, severely degraded images, and visually non-dominant instances, whose embeddings are more likely to be unstable and less faithful to the class manifold. By incorporating $\mathcal{L}_{\mathrm{conf}}$ into the subset objective optimized by greedy construction and swap-based refinement, such low-confidence candidates become less likely to be selected and more likely to be replaced if selected, thereby preventing unreliable geometric anchors from entering $S_c$. This interpretation is consistent with our ablation behavior: removing $\mathcal{L}_{\mathrm{conf}}$ primarily decreases IS, indicating degraded class-conditional semantic clarity due to the inclusion of semantically unreliable selections.

# H. More Visualization Results

Figure 8 showcases $512 \times 512$ images generated by DiT-L/2 trained for only 10K iterations on the condensed dataset obtained with our method. The results demonstrate high visual fidelity, rich diversity, and rapid convergence, indicating that our condensed subset provides broad and faithful coverage of the real data manifold, effectively supporting high-quality diffusion training even under limited iterations. Figure 9 further presents $256 \times 256$ samples generated by DiT-L/2 trained

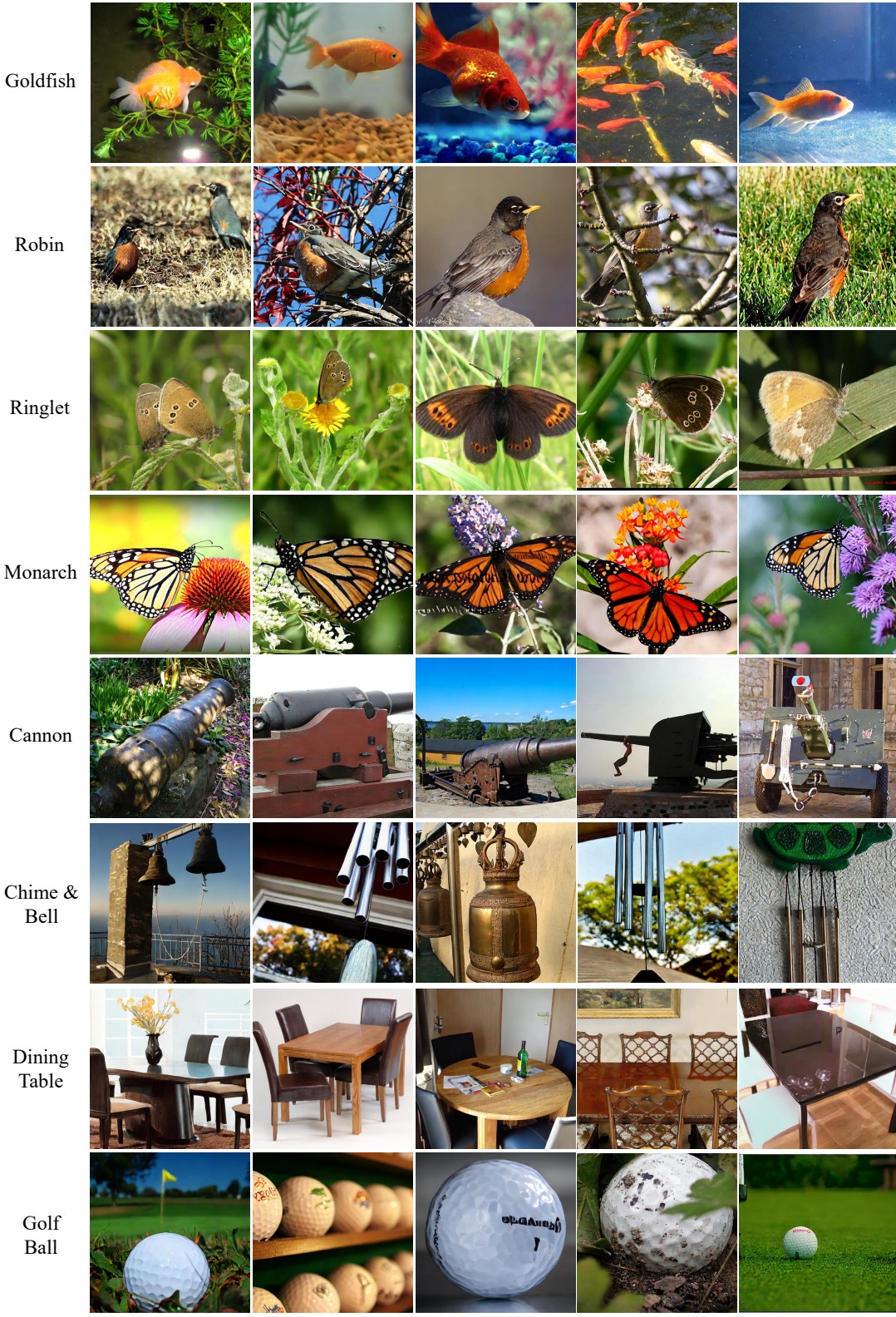

*Figure 8.* Examples of $512 \times 512$ images generated by DiT-L/2 (Peebles & Xie, 2023) after 10K training iterations on a condensed dataset (10K data budget) selected by our method.

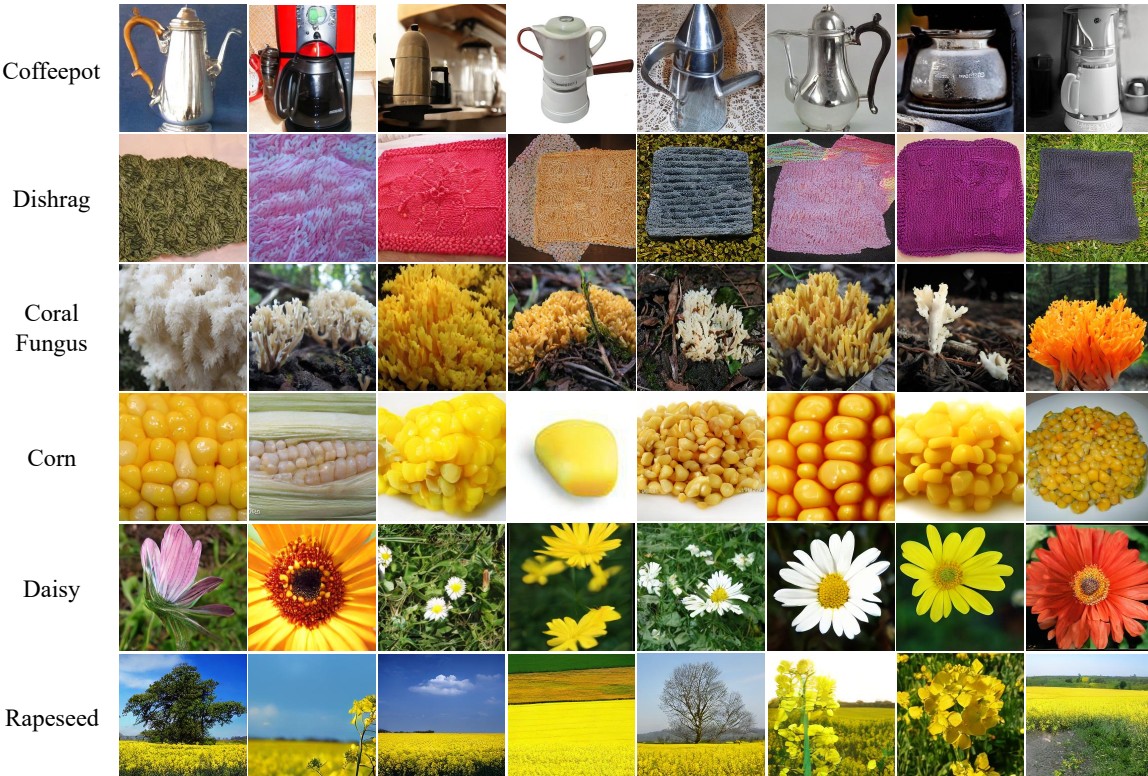

*Figure 9.* Examples of $256 \times 256$ images generated by DiT-L/2 (Peebles & Xie, 2023) after 30K training iterations on a condensed dataset (50K data budget) selected by our method.

for 30K iterations using our 50K-image condensed subset. The model produces diverse and semantically coherent images with well-preserved fine-grained details, illustrating that the larger budget enhances coverage of relatively low-density regions and yields stable generative behavior.

Figure 7 provides a qualitative comparison between $D^2C$ (Huang et al., 2026) and our method on a representative ImageNet class (*tench*). We visualize the real data distribution in the embedding space using t-SNE, where gray dots denote all real samples and blue markers indicate the selected subset. As shown in Figure 7 (left), $D^2C$ selects samples by uniformly spacing along a scalar difficulty ranking, which can lead to uneven coverage of the underlying embedding distribution and leave certain regions under-represented. In contrast, our method (Figure 7, right), guided by one-sided partial optimal transport, selects samples that are more broadly distributed across the data manifold. This results in higher coverage and a reduced mean nearest-neighbor distance to the real data, providing qualitative evidence that our approach achieves better geometric alignment with the full data distribution.

## I. PseudoCode

We present the pseudocode for our condensation pipeline in Algorithm 1. The computation of the one-sided partial optimal transport objective is detailed in Algorithm 2, while the greedy initialization and the subsequent swap-based refinement procedures are described in Algorithm 3 and Algorithm 4, respectively.

At each greedy or swap evaluation, we compute the composite objective $\mathcal{L}(\mathcal{S}_c, \mathcal{T}_c) = \mathcal{L}_{\text{OT}} + \alpha \, \mathcal{L}_{\text{stat}} + \beta \, \mathcal{L}_{\text{conf}}$, where $\mathcal{L}_{\text{OT}}$ is obtained by invoking the Sinkhorn solver in Algorithm 2 using the current condensed embeddings $\mathcal{S}_{c(e)}$ and the fixed real embeddings $\mathcal{T}_{c(e)}$. In practice, for a fixed $x_i$ we evaluate all candidate swaps $(x_i \to x_j)$ in a batched manner by recomputing $\mathcal{L}_{\text{OT}}$ with the updated $\mathcal{S}'_{c(e)}$, enabling GPU-parallel scoring of $\Delta_{i \to j}$.

---

**Algorithm 1** Per-Class Alignment-driven Real Subset Selection via One-Sided Partial OT

---

**Require:** Full dataset $\mathcal{T} = \bigcup_{c=1}^{C} \mathcal{T}_c$; subset size per class $m$; embedding extractor (to obtain $\mathcal{T}_{c(e)}$); entropic OT parameters $(\varepsilon, T, \kappa)$; weights $(\alpha, \beta)$; batch size $B$; maximum refinement iterations $I$

**Ensure:** Condensed subset $\mathcal{S} = \bigcup_{c=1}^{C} \mathcal{S}_c$ with $|\mathcal{S}_c| = m$

1: **for** each class $c$ **do**
2:     Compute feature embeddings $\mathcal{T}_{c(e)} = \{\mathbf{y}_j\}_{j=1}^{n}$ from $\mathcal{T}_c$
3:     Compute confidence scores $p(c \mid \mathbf{y}_j)$ for $\mathcal{T}_c$
4:     Initialize $\mathcal{S}_c \leftarrow \text{GREEDYINIT}(\mathcal{T}_{c(e)}, m)$
5:     Refine $\mathcal{S}_c \leftarrow \text{SWAPREFINE}(\mathcal{S}_c, \mathcal{T}_{c(e)}, I)$
6: **end for**
7: Gather $\{\mathcal{S}_c\}$ and return $\mathcal{S}$

---

**Algorithm 2** One-Sided Partial OT Loss $\mathcal{L}_{\text{OT}}$ via Dummy Source

---

**Require:** Condensed embeddings $\mathcal{S}_{c(e)} = \{\mathbf{x}_i\}_{i=1}^{m}$; real embeddings $\mathcal{T}_{c(e)} = \{\mathbf{y}_j\}_{j=1}^{n}$; $\varepsilon$; total Sinkhorn iters $T$; capacity scaling $\kappa \geq 1$; and scaling coefficient $\gamma$

**Ensure:** One-sided partial OT loss $\mathcal{L}_{\text{OT}}$

1: Construct cost matrix $\mathbf{C} \in \mathbb{R}^{m \times n}$ with $\mathbf{C}_{ij} = \|\mathbf{x}_i - \mathbf{y}_j\|_2^2$

2: Construct augmented cost matrix $\mathbf{C}_{\text{aug}} = \begin{bmatrix} \mathbf{C} \\ \text{median}(\mathbf{C})\gamma\mathbf{1}_n^\top \end{bmatrix}$

3: Set source marginal $\boldsymbol{\mu} = \frac{1}{m}\mathbf{1}_m$ {fully shipped}
4: Set reference target marginal $\bar{\boldsymbol{\nu}} = \frac{1}{n}\mathbf{1}_n$ and target capacity $\boldsymbol{\nu} = \kappa\bar{\boldsymbol{\nu}}$
5: Set dummy mass $s = \kappa - 1$ and augmented source marginal $[\boldsymbol{\mu}; s]$
6: Initialize Gibbs kernel $\mathbf{K} = \exp(-\mathbf{C}_{\text{aug}}/\varepsilon)$
7: Initialize Sinkhorn scalings $\mathbf{u} \leftarrow \mathbf{1}, \mathbf{v} \leftarrow \mathbf{1}$
8: **for** $r = 1$ to $T$ **do**
9:     $\mathbf{u} \leftarrow [\boldsymbol{\mu}; s] \oslash (\mathbf{Kv})$
10:    $\mathbf{v} \leftarrow \boldsymbol{\nu} \oslash (\mathbf{K}^\top \mathbf{u})$
11: **end for**
12: Compute augmented transport plan $\boldsymbol{\Pi} = \text{Diag}(\mathbf{u})\,\mathbf{K}\,\text{Diag}(\mathbf{v})$
13: Remove dummy row to obtain $\mathbf{T}_{\text{real}} \leftarrow \boldsymbol{\Pi}_{1:m,:}$
14: $\mathcal{L}_{\text{OT}} \leftarrow \langle \mathbf{C}, \mathbf{T}_{\text{real}} \rangle$
15: **return** $\mathcal{L}_{\text{OT}}$

---

## J. Broader Impact

Our geometry-aware dataset condensation framework enhances the efficiency of diffusion model training by reducing data size while preserving structural fidelity, enabling sustainable and accessible generative modeling. This advancement holds potential for widespread use in text-to-image generation, multimodal learning, and data-centric optimization, lowering the carbon footprint and democratizing large-scale generative modeling. Future work can further incorporate GPT-based evaluation as a complementary assessment protocol (Cui et al., 2024c; Sun et al., 2024b) and explore its use for data augmentation (Lv & Jiang, 2026). Nevertheless, dataset condensation inherently modifies the data distribution and may inadvertently amplify biases or underrepresent minority patterns if applied without care. Furthermore, condensed subsets preserve critical structural and semantic information, raising concerns about privacy, copyright, and potential misuse for generating synthetic or misleading media. These risks can be mitigated through fairness auditing, balanced sampling, anonymization, and transparent documentation of data provenance and condensation settings. Overall, our method advances energy-efficient AI while underscoring the importance of responsible governance, ethical safeguards, and open evaluation to ensure that data efficiency and social accountability progress hand in hand.

---

**Algorithm 3** Stage I: Greedy Initialization for $\mathcal{S}_c$

---

**Require:** Real set $\mathcal{T}_c$; real embeddings $\mathcal{T}_{c(e)} = \{\mathbf{y}_j\}_{j=1}^n$; subset size $m$
**Ensure:** Initial condensed subset $\mathcal{S}_c$ with $|\mathcal{S}_c| = m$
  1: $\mathcal{S}_c \leftarrow \emptyset$
  2: $x_1^\star \leftarrow \arg\min_{x_k \in \mathcal{T}_c} \mathcal{L}(\{x_k\}, \mathcal{T}_c)$
  3: $\mathcal{S}_c \leftarrow \{x_1^\star\}$
  4: **for** $t = 2$ to $m$ **do**
  5:     For each candidate $x_k \in \mathcal{T}_c \setminus \mathcal{S}_c$, compute marginal gain
  6:         $\Delta\mathcal{L}(x_k) \leftarrow \mathcal{L}(\mathcal{S}_c \cup \{x_k\}, \mathcal{T}_c) - \mathcal{L}(\mathcal{S}_c, \mathcal{T}_c)$
  7:     $x_t^\star \leftarrow \arg\min_{x_k \in \mathcal{T}_c \setminus \mathcal{S}_c} \Delta\mathcal{L}(x_k)$
  8:     $\mathcal{S}_c \leftarrow \mathcal{S}_c \cup \{x_t^\star\}$
  9: **end for**
 10: **return** $\mathcal{S}_c$

---

**Algorithm 4** Stage II: Swap-Based Refinement for $\mathcal{S}_c$

---

**Require:** Current subset $\mathcal{S}_c$ with $|\mathcal{S}_c| = m$; real set $\mathcal{T}_c$; max rounds $I$
**Ensure:** Refined subset $\mathcal{S}_c$
  1: **for** $t = 1$ to $I$ **do**
  2:     improved $\leftarrow$ **false**
  3:     **for all** $x_i \in \mathcal{S}_c$ **in fixed order do**
  4:         Evaluate $\Delta_{i \to j}$ for all $x_j \in \mathcal{T}_c \setminus \mathcal{S}_c$ in parallel
  5:         $j^\star \leftarrow \arg\min_{x_j \in \mathcal{T}_c \setminus \mathcal{S}_c} \Delta_{i \to j}; \quad \Delta^\star \leftarrow \Delta_{i \to j^\star}$
  6:         **if** $\Delta^\star < 0$ **then**
  7:             $\mathcal{S}_c \leftarrow (\mathcal{S}_c \setminus \{x_i\}) \cup \{x_{j^\star}\}$
  8:             improved $\leftarrow$ **true**
  9:         **end if**
 10:     **end for**
 11:     **if not** improved **then**
 12:         **break**
 13:     **end if**
 14: **end for**
 15: **return** $\mathcal{S}_c$

---

