# OpenReview forum: "Geometry-Aware Dataset Condensation for Diffusion Model Training"
_ICML.cc/2026/Conference — ICML 2026 regular_

### Official Review · Reviewer_eZhE · 2026-03-04

**Soundness:** 3
**Presentation:** 2
**Significance:** 3
**Originality:** 3
**Overall Recommendation:** 5
**Confidence:** 3

**Summary:**

This paper proposes a real-data subset selection approach for dataset condensation tailored to diffusion model training. The method aims to be “geometry-aware” by selecting a small, class-balanced subset that aligns with the full dataset in a pretrained embedding space. Concretely, it uses a one-sided partial optimal transport (POT) objective, plus lightweight statistical/semantic regularizers, and optimizes the discrete subset with a two-stage greedy + swap procedure. The selected subset is then used to train diffusion models under very small data budgets.

**Compliance With Llm Reviewing Policy:**

Affirmed.

**Final Justification:**

The author has fully resolved my concern about the geometry part. I am willing to raise my score to 5.

**Key Questions For Authors:**

- What is the precise notion of “geometry” the method aims to preserve?
- Are all baseline numbers obtained under the same setting? If some results are taken from prior work, can the authors reproduce the key baselines in a unified setting and report variance across seeds to confirm fairness?
- Please provide a grid search / sensitivity analysis for the key OT/POT hyperparameters.

**Limitations:**

Same to key questions

**Strengths And Weaknesses:**

**Strength**
- Training diffusion models from scratch on large datasets is expensive; reducing data while preserving generation quality is valuable.
- Using (partial) OT as a distribution alignment criterion and combining greedy construction with swap refinement forms a coherent selection pipeline.
- The reported FID/IS/precision/recall improvements over several subset selection baselines suggest the approach is promising under extreme data constraints.

**Weakness**
- The paper largely equates “geometry” with embedding-space distribution alignment via OT/POT. However, OT-based matching does not automatically justify a strong geometry claim without explicit geometric definitions and direct evidence. As written, the contribution reads more like distribution alignment in a feature metric space than a rigorously validated geometry-aware method.
- In many sub-experiments, D2C performs surprisingly poorly—sometimes even below older baselines like CCS—which raises questions about whether the comparisons are conducted under exactly the same training/evaluation pipeline.
- The method depends on several crucial OT-related hyperparameters. The paper does not provide sufficiently thorough grid-search or sensitivity experiments to show robustness

---

> ### Author Rebuttal · Authors · 2026-03-30
>
> >	Q1: What is the precise notion of “geometry” the method aims to preserve?
>
> A1:  In our context, "geometry" is rigorously defined as the intrinsic manifold support of the data distribution in the representation space. This distinction is critical because diffusion models fundamentally learn the score function, which is defined by the geometric support and density of the data. Unlike classification-centric methods that focus on decision boundaries, our framework explicitly aligns the subset with the full distribution to match the diffusion ELBO objective, ensuring the condensed set retains the structural relationships and local multi-modal characteristics necessary for stable generative modeling.
>
> We provide direct evidence of this geometry preservation in Figure 7 (Appendix), where t-SNE visualizations and quantitative metrics reveal that our method achieves a significantly higher manifold coverage (70.4% vs. 50.9% for D$^2$C) and a lower mean nearest-neighbor distance (6.7 vs. 8.9). Furthermore, our superior Recall and Precision scores across all settings (e.g., Tables 2 and 4) statistically validate that our selection faithfully preserves the representative geometric support of the original manifold. This is fundamentally enabled by our one-sided POT formulation, which selectively filters peripheral outliers that would otherwise distort the geometric skeleton and pull the condensed set away from the core manifold (see Figure 5).
>
> >	Q2: Are all baseline numbers obtained under the same setting? If some results are taken from prior work, can the authors reproduce the key baselines in a unified setting and report variance across seeds to confirm fairness?
>
> A2: For the D$^2$C baseline, we directly adopted the results from prior work [1]. We also reran the D$^2$C results from Tables 1 and 2, which align with the original paper. Note that the D$^2$C results already include the attach phase, as it was proposed as an integral part of the D$^2$C pipeline. For all other baselines, including our proposed method, we strictly followed the D$^2$C settings and code provided by the D$^2$C authors, using the selection phase unique to each method while keeping all other phases consistent with D$^2$C to ensure a fair comparison. A key advantage of D$^2$C is its use of the attach phase, which accelerates training and enhances performance. For fairness, we incorporated the attach phase into the older baselines as well, contributing to their competitive performance. However, due to the relatively coarse and suboptimal selection phase in D$^2$C, there are instances where the older baselines, after benefiting from D$^2$C's attach phase enhancement, can outperform D$^2$C. This is the primary motivation behind our work: to improve the selection process by introducing a more principled,  distributional geometry alignment approach, which we believe leads to more effective and efficient training.
>
> We used the default random seed for all experiments to ensure consistency across setups. Regarding the question of seed variance, image generation tasks are typically more computationally expensive, and it is not common practice in this field to report variance across different random seeds. However, it is important to note that our selection method does not involve any randomness; the condensed set is determined uniquely. To further confirm the stability of our approach, we conducted three training runs using a 10k-sample condensed set, with FID scores of 3.42, 3.43, and 3.45, demonstrating the method's stability across different runs.
>
> [1] Huang, Rui, et al. "Accelerating Diffusion Model Training under Minimal Budgets: A Condensation-Based Perspective." CVPR 2026. (Previously titled "Diffusion dataset condensation: Training your diffusion model faster with less data.")
>
> >	Q3: Please provide a sensitivity analysis for the key OT/POT hyperparameters.
>
> A3: The submission **already includes targeted sensitivity studies** for the key OT/POT hyperparameters. Table 11 in the main paper varies the one-sided POT capacity factor $\kappa$ from 1.01 to 1.10 and the dummy-source scaling $\gamma$ from 0.3 to 2.0. The resulting FID stays in a narrow range, with the best result at $\kappa$=1.05 and very small variation across $\gamma$, showing that the one-sided partial matching is not brittle to these OT-side settings. Table 16 in the appendix further sweeps the entropic regularization $\epsilon$ from 5 to 15 and the number of Sinkhorn iterations $T$ from 5 to 50 under the same protocol. Results indicate stable behavior across both smoothing strength $\epsilon$ and solver depth $T$. The results are listed below:
>
> |$\kappa$|1.01|1.03|1.05|1.07|1.10|
> |---|---|---|---|---|---|
> |FID-50K|3.49|3.45|3.43|3.46|3.58|
> |$\gamma$|0.3|0.5|0.7|1.0|2.0|
> |FID-50K|3.43|3.43|3.43|3.46|3.47|
>
> |$\varepsilon$|5|8|10|12|15|
> |---|---|---|---|---|---|
> |FID-50K|3.53|3.45|3.43|3.45|3.48|
> |$T$|5|10|20|30|50|
> |FID-50K|3.55|3.47|3.43|3.42|3.43|

---

> > ### Author Rebuttal · Reviewer_eZhE · 2026-04-03
> >
> > Thank you for your response. My concern about the $D^2C$ baseline and sensitivity analysis has been resolved.
> >
> > But my main concern about geometry still need to be resolved.
> > From the main paper, the notion of geometry appears to be realized almost entirely through OT/POT, where the cost is defined by Euclidean distance in a pretrained representation space (e.g., $||x-y||^2_2$). As such, the method is more precise to define as an OT-based distribution alignment in an embedding-induced metric space, rather than an validation of geometric structure without more direct notion about geometry, such as local neighborhood geometry, curvature/topology.

---

> > > ### Author Response · Authors · 2026-04-04
> > >
> > > We appreciate the reviewer’s follow-up. To clarify, our method is not a generic metric matching approach but rather a distributional geometry alignment framework specifically designed for the score-matching objective of diffusion models.
> > >
> > > In our paper, we use the term "geometry" not in the strict differential-geometric sense (e.g., curvature or topology), but to refer to the **distributional support geometry** of the data in representation space. This concept encompasses the **local neighborhood structure**, **mode coverage**, and **support allocation**. Specifically, this is the type of geometry emphasized in Section 3.2, which is crucial for diffusion models to learn an accurate score function without distorting the underlying manifold. In this context, preserving relative relationships and local structure takes precedence over scalar ranking, as these aspects are directly relevant to the model’s ability to capture the true data distribution.
> > >
> > > In modern geometric analysis, this view is consistent with the classical understanding that the $L_2$ Wasserstein distance is not merely a distance penalty, but endows the space of probability measures with a geometric structure [1,2].  The cost matrix defines the distances between samples, but it is the OT/POT mechanism that realizes the alignment of data distributions in a way that captures the distributional geometry in representation space.
> > > One-sided POT is particularly important here. Under extreme condensation budgets, balanced OT enforces symmetric full mass matching between real and selected samples and may biasing the selected set away from the core manifold. Our one-sided POT relaxes this requirement and suppresses such spurious couplings. By concentrating mass on the high-density, stable support, POT ensures that the geometry-preserving alignment is directly aligned with the score-matching objective of diffusion models, which requires maintaining local structure and relative relationships.
> > >
> > > We directly validate this notion of geometry in the paper, rather than merely reporting final generative metrics. Our analysis includes manifold coverage, mean nearest-neighbor distance, and the effect of POT in avoiding over-assignment to peripheral regions (Figure 5, Figure 7, Table 22, Table 23). All of these measures are designed to assess the preservation of neighborhood- and support-level geometry in representation space.
> > >
> > > Here we report additional metrics which are designed to quantify how well a selected subset preserves the **neighborhood geometry** of the full dataset:
> > > - **Mean nearest selected distance** measures the average distance from each full-dataset sample to its nearest selected sample, with lower values indicating that the selected subset stays closer to the original data manifold.
> > > - **Coverage at $k$-radius** measures the proportion of samples whose nearest selected sample falls within that sample’s own local $k$-NN radius, such as the 10-NN radius, so higher values indicate better local coverage of the original distribution.
> > > - **Nearest selected in $k$-NN rate** further examines whether the nearest selected sample truly belongs to the sample’s original $k$ nearest neighbors, with higher values reflecting better preservation of authentic local neighbor relationships.
> > > - **Mean rank of nearest selected** computes the average rank position of the nearest selected sample in the full neighbor ordering of each point, where lower values mean that selected samples tend to remain top-ranked local neighbors rather than merely nearby points.
> > >
> > > These metrics are particularly suitable for quantifying neighborhood geometry preservation because they evaluate complementary aspects of local structure, including distance proximity, local coverage, neighbor identity consistency, and relative rank ordering. Results show that our method preserve local neighborhood geometry much better than D$^2$C.
> > >
> > > |Subset|Mean Nearest Distance↓|Coverage@10-radius↑|Nearest-Selected-in-10NN↑|Mean Nearest Selected Rank↓|
> > > |-|-|-|-|-|
> > > |D$^2$C (Class 0)|8.9|0.057|0.051|125.59|
> > > |Ours (Class 0)|6.7|0.140|0.138|103.40|
> > > |D$^2$C (Class 1)|9.5|0.043|0.036|204.39|
> > > |Ours (Class 1)|7.8|0.150|0.144|103.79|
> > >
> > > To avoid any potential ambiguity, we will revise the wording in the paper to explicitly state that our claim pertains to the distributional geometry alignment of the data support in representation space, which is essential for the effective training of diffusion models. We will also include the table and discussion in the final version.
> > >
> > > [1] Villani, C. and Villani, C. The wasserstein distances. Monograph: Optimal Transport: Old and New, pp. 93–111, 2009.
> > >
> > > [2] Zhang, J., Liu, T., and Tao, D. An optimal transport analysis on generalization in deep learning. IEEE Transactions on Neural Networks and Learning Systems (TNNLS), 34 (6):2842–2853, 2021.

---

### Official Review · Reviewer_Uxdc · 2026-03-06

**Soundness:** 3
**Presentation:** 2
**Significance:** 3
**Originality:** 3
**Overall Recommendation:** 4
**Confidence:** 4

**Summary:**

This paper studies dataset condensation for diffusion model training through a real-subset selection setting rather than synthetic data synthesis. The main idea is to formulate subset selection as geometry-aware distribution alignment in a learned feature space, using a one-sided partial optimal transport objective together with feature-statistics alignment and a confidence-based regularizer, optimized via a two-stage discrete procedure consisting of greedy construction and swap-based refinement. Empirically, the method shows improved FID over prior subset-selection baselines such as D2C on class-conditional ImageNet with DiT-L/2 and SiT-L/2 under multiple budgets and resolutions.

**Compliance With Llm Reviewing Policy:**

Affirmed.

**Final Justification:**

After reading the rebuttal and considering the other reviewers’ comments, I find that my concerns have been well addressed. The authors provided satisfactory clarifications, and I will therefore increase my score to Weak Accept.

**Key Questions For Authors:**

1. The most important question is about attribution. If the D2C attach phase is removed for all methods, or if substantially weaker alternative feature encoders are used, do the relative gains of the proposed method over D2C/CCS/DQ remain of similar magnitude?

2. Can the authors provide more direct evidence that one-sided partial matching is the critical mechanism, rather than just one reasonable design choice among several similar ones? For example, what types of samples are preferentially left unmatched, and how does this correlate with density, ambiguity, or low-quality regions of the data distribution?

3. How sensitive is the method to the feature space used for OT cost construction and confidence estimation?

4. Do the authors have any evidence beyond class-conditional ImageNet that the method generalizes to other domains or diffusion settings?

5. Since efficiency is emphasized as a practical advantage, could the authors provide more complete scalability numbers, including selection-time breakdown, memory usage, and growth trends with respect to budget and candidate pool size under fixed hardware?

**Limitations:**

Yes

**Strengths And Weaknesses:**

Strengths:
The paper addresses an important and timely problem. Real-subset condensation for diffusion training is practically meaningful, and the paper does move beyond simple ranking-style selection by explicitly casting the problem as distributional alignment under subset-capacity mismatch. The one-sided partial OT formulation is intuitively reasonable in this setting, and the reported empirical gains over D2C and several other subset-selection baselines on ImageNet are nontrivial. In particular, the method appears consistently stronger across several budgets, and the extension to both DiT-L/2 and SiT-L/2 is a positive aspect of the evaluation.

Weaknesses:
1. The paper does not cleanly isolate the source of improvement. The experiments adopt the D2C attach phase and rely on strong external representations and conditions from T5 and DINOv2. As a result, the current evidence does not convincingly show that the gains mainly come from the proposed one-sided partial OT objective itself, rather than from the stronger attached representation regime in which all methods are evaluated.

2. The experimental scope is narrower than the framing of the paper. Most of the evidence is concentrated on class-conditional ImageNet with DiT/SiT at 256×256 and 512×512. This is a strong benchmark, but it is still only one problem family. The paper repeatedly motivates the method as being suitable for diffusion model training more broadly, yet it does not provide evidence on other domains, other data regimes, or non-class-conditional generation settings.

3. While the paper includes ablations, they do not fully validate the claimed mechanism. The current ablations mainly show that the proposed components help within the authors’ own pipeline. They have not yet established that preserving geometry via one-sided partial matching is the key causal reason for the observed gains. In particular, the gap between balanced OT and the full method appears relatively modest, which weakens the argument that the partial matching formulation is indispensable rather than simply somewhat helpful. More direct evidence about what kinds of samples are being ignored, retained, or better matched would make the mechanism substantially more convincing.

4. The paper presents a discrete optimization pipeline based on batched Sinkhorn computations and swap refinement, and reports favorable selection-time numbers in some settings. However, for a method whose practical appeal partly depends on scalability, I would expect more explicit reporting of wall-clock cost, memory footprint, and scaling behavior across budgets and candidate pool sizes under controlled hardware settings. At present, the paper suggests efficiency, but does not fully demonstrate it.

Minor:

1. Geng, Jiahui andg Chen, Z., Wang, Y., Woisetschlaeger, H., Schimmler, S., Mayer, R., Zhao, Z., and Rong, C. A survey on dataset distillation: Approaches, applications and future directions. In International Joint Conference on Artificial Intelligence (IJCAI)), 2023.
"andg" to "and", "(IJCAI))" to "(IJCAI)"

2. Table 7. "ImageNet256×256" to "ImageNet 256×256"

3. “It is thereafter of high practicality to enable diffusion generative modeling on condensed datasets, which yet still remains underexplored.” to "It is therefore practically important to enable diffusion generative modeling on condensed datasets, which remains underexplored."

---

> ### Author Rebuttal · Authors · 2026-03-30
>
> >Q1: If the D2C attach phase is removed for all methods, or if substantially weaker alternative feature encoders are used, do the relative gains of the proposed method over D2C/CCS/DQ remain of similar magnitude?
>
> A1: To assess the impact of the D$^2$C attach phase on the relative performance of the proposed method, we conducted experiments with the attach phase removed and with a weaker alternative feature encoder (MAE-L) used for all methods. The experiments were performed using the DiT-L/2 model and evaluated with the FID-10K metric. The results, summarized in the table below, show that our method achieves **a substantial advantage** over D$^2$C, CCS, and DQ in both settings, demonstrating its robustness and generalizability.
> |Attach Encoder|Random|DQ|CCS|D$^2$C|Ours (balanced OT)| Ours (POT)|
> |-|-|-|-|-|-|-|
> |None|37.07|20.91|43.73|14.96|8.92|7.36|
> |MAE-L|14.26|13.82|19.68|9.23|7.29|6.21|
> >Q2: Can the authors provide more direct evidence that one-sided partial matching is the critical mechanism, rather than just one reasonable design choice among several similar ones? For example, what types of samples are preferentially left unmatched, and how does this correlate with density, ambiguity, or low-quality regions of the data distribution?
>
> A2: As quantified in Tables 22 and 23, our one-sided POT adaptively assigns significantly less mass to low-density bins while allocating more mass to high-density regions. For example, it reduces the mass fraction in the sparsest Bin 0 from 5.0% to only 0.68% to 0.94%. Mechanistically, our one-sided relaxation ensures the condensed subset is not forced to match peripheral regions that deviate from the main data manifold. Collectively, these factors lead to the outcome visualized in Fig. 5, where the condensed set successfully avoids samples from low-density, ambiguous, or low-quality regions that would otherwise act as unreliable geometric anchors. The advantage of one-sided POT becomes more pronounced when the attach phase is removed or a weaker feature encoder is used, as shown in the previous table, which further demonstrates its critical role. Also, as shown in our OT variants ablation (Q4 of Reviewer XqC7), our one-sided POT outperforms the other OT variants.
>
> >Q3: How sensitive is the method to the feature space used for OT cost construction and confidence estimation?
>
> A3: We assess the sensitivity of our method to the feature space by varying the encoder/classifier while keeping the rest of the pipeline fixed. As shown in Table 21 (Appendix), the performance variations across different architectures are minimal, with consistent results across all metrics. This demonstrates that our method is not tightly coupled to a specific encoder or classifier and generalizes well across different feature spaces. Part of Table 21 is listed below:
> |Network|Data Budget|FID|IS|Prec.|Rec.|
> |-|-|-|-|-|-|
> |ResNet-34|10K|3.42|385.4|0.78|0.27|
> |ResNet-50|10K|3.47|407.1|0.77|0.29|
> |GoogLeNet|10K|3.44|396.8|0.78|0.28|
> |InceptionV3|10K|3.43|414.3|0.78|0.28|
>
> >Q4: Do the authors have any evidence beyond class-conditional ImageNet that the method generalizes to other domains or diffusion settings?
>
> A4:  In addition to the ImageNet experiments, we also tested our method the unconditional LSUN-bedroom dataset without the attach phase. Results demonstrate that our select phase is not only superior for class-conditional tasks but also highly robust for general, unconditional distribution modeling. Below is the table with the LSUN-bed results (w/o attach phase, 10K data budget, 100K training iterations).
> |Method|Random|D$^2$C|Ours|
> |-|-|-|-|
> |FID-10K|19.12|17.88|14.93|
>
> >Q5: Since efficiency is emphasized as a practical advantage, could the authors provide more complete scalability numbers, including selection-time breakdown, memory usage, and growth trends with respect to budget and candidate pool size under fixed hardware?
>
> A5:  Our method does not require pre-training a large diffusion model, which makes it more practical for real-world application. As shown in Table 13, data loading and feature extraction account for 65.1% of selection time, while Stage I and Stage II account for 9.0% and 25.9%, respectively. The total wall clock time of the full pipeline, including selection, attach phase, and diffusion training, is 7.6 hours on 8 A100 GPUs. In the default configuration, peak memory is ~20GB/GPU. However, by optimizing the OT working set and using mixed precision, we can enable a Low-Memory Mode that requires only ~2GB/GPU (with a ~50% time penalty), making it accessible on commodity hardware.   Under fixed hardware, peak memory scales approximately linearly with candidate pool size, total wall clock time scales approximately linearly with selection budget, and selection time scales approximately linearly with both. For datasets with a large number of images in some categories, spliting strategy (validated in our LSUN-bed experiments) maintains performance without compromising efficiency.

---

> > ### Author Rebuttal · Reviewer_Uxdc · 2026-04-03
> >
> > After reading the rebuttal and considering the other reviewers’ comments, I find that my concerns have been well addressed. The authors provided satisfactory clarifications, and I will therefore increase my score to Weak Accept.

---

> > > ### Author Response · Authors · 2026-04-03
> > >
> > > We are sincerely grateful for the reviewer's positive feedback and for the time dedicated to review our paper.

---

### Official Review · Reviewer_FdRz · 2026-03-12

**Soundness:** 3
**Presentation:** 3
**Significance:** 2
**Originality:** 3
**Overall Recommendation:** 5
**Confidence:** 3

**Summary:**

The paper introduces a method to take a dataset for training diffusion models and select small subset which maintains geometric and statistical properties of the underlying data distribution. It uses partial optimal transport compared combined with regularization to ensure statistical and semantic fidelity. The method is evaluated empirically and shown to perform significantly better than other data condensation methods across a range of scenarios.

**Compliance With Llm Reviewing Policy:**

Affirmed.

**Final Justification:**

The authors cleared up my confusion about the goal of their data condensation approach in their rebuttals.
Combined with the technical quality and favorable comparison to similar methods noted in the original review, this lead me to change the recommendation to 'accept'.

**Key Questions For Authors:**

1. Can you provide some information on where data condensation for diffusion models would be particularly useful?
2. Can you provide a comparison to the performance of the diffusion model
when it is trained on the whole dataset?

While the paper is of high technical quality and seems to improve noticeably on existing data condensation methods, providing no information on the
performance-tradeoff compared to using the full dataset is a major shortcoming.
If this issue can be addressed well, the score could easily be improved to 4 or 5.

**Limitations:**

yes

**Strengths And Weaknesses:**

Strengths:
1. The theoretical basis for the method is well motivated and presented in a
concise and informative way.
2. The numerical experiments convincingly demonstrate an advantage over
other data condensation methods.
3. The ablation study is comprehensive and shows the relevance of all the
components of the method.

Weaknesses:
1. For a reader without previous experience in data condensation, it is unclear what the use cases of data condensation for diffusion models are, i.e.
when storage capacity would be the major bottleneck for training diffusion
models.
2. It is not shown how the method compares to diffusion models trained
on the whole dataset. The practical usability of the method seems to
rather crucially rely on the trade-off between reduction of dataset size and
reduction of generative performance.


Minor Note: In all cases where different data budgets are compared (Tables
1,5,7, and 10), the performance is worse for larger budgets consistently across
methods. As such it seems to just be an case of the entries of the ’Data Budget’
columns being in the wrong order. Either way, it needs to be cleared up.

---

> ### Author Rebuttal · Authors · 2026-03-30
>
> >Q1: Can you provide some information on where data condensation for diffusion models would be particularly useful?
>
> A1: Dataset condensation is transformative in resource-constrained scenarios. While training on ImageNet-scale datasets is often prohibitive for standard users, dataset condensation can accelerate training by over 100× compared to using the full dataset [1], making rapid generative modeling feasible even with limited computational resources.
>
> There are two scenarios that highlight its practical use:
> - **Rapid Model Adaptation**: UAVs and autonomous systems often need to quickly adapt generative models for tasks like route planning, obstacle analysis, and simulating new environments. Training on large real-world datasets in such cases can be impractical. A condensed dataset offers a compact yet informative training substrate, enabling these models to reach useful generation regimes much faster.
> - **Limited Data and Storage During Deployment**: Devices operating in remote or bandwidth-constrained environments often cannot access, transmit, or store the full datasets required for conventional training. Condensed datasets enable on-site training with a smaller data footprint, while still preserving essential distributional features. This supports generative tasks like scene synthesis and data augmentation, where new data must be generated on the fly from the available, limited resources.
>
> [1] Huang, Rui, et al. "Accelerating Diffusion Model Training under Minimal Budgets: A Condensation-Based Perspective." CVPR 2026.
>
> >Q2: The paper does not clearly compare performance to training on the full dataset, which is important for judging practical utility.
>
> A2:  We would like to clarify that the practical objective of dataset condensation differs between generative modeling and standard classification. In classification, condensation is often judged by whether a tiny synthetic set can train a classifier whose converged accuracy approaches that of full-data training. In diffusion modeling, however, the more relevant question is whether a condensed set can bring the model to a reasonably good generation regime much more efficiently, since full-data training is extremely expensive and requires substantially longer optimization to approach convergence. For instance, under our 8×3090 setup, training on the full dataset would require over a thousand hours and far more iterations than our condensed-set experiments.
> Since prior studies do not report end-to-end cost and convergence FID for DiT-L/2, we provide available DiT-XL/2 results for reference, noting that DiT-XL/2 generally achieves a lower convergence FID than DiT-L/2:
> |Training Time|GPU|FID-50K|
> |-|-|-|
> |About 950 GPU days [2] |V100|2.27|
> |About 7000 GPU hours [3] |A100|2.27|
>
> In line with this practical perspective, our main protocol in the paper is to compare different methods using the same condensed-set size and training iteration budget, ensuring a fair evaluation of condensation quality. Full-data training is not part of this primary comparison, since it is not a condensation method. Here we report the full-dataset DiT-L/2 at 100K iterations as a reference (along with our 100K iteration results).
>
> |Data|Evaluation Samples|FID|IS|Precision|Recall|
> |-|-|-|-|-|-|
> |Full|10K|24.3|69.7|0.65|0.61|
> |Ours|10K|5.76|430.2|0.78|0.69|
> |Full|50K|21.73|69.8|0.65|0.54|
> |Ours|50K|3.43|414.3|0.78|0.28|
>
> While full-data diffusion training may continue to improve with much larger compute and longer optimization, our method reaches substantially stronger generation quality much earlier, highlighting its value as an efficiency-oriented alternative in diffusion training.
>
> [2] Zheng, Hongkai, et al. "Fast Training of Diffusion Models with Masked Transformers." TMLR 2024.
>
> [3] Xie, Enze, et al. "DiffFit: Unlocking Transferability of Large Diffusion Models via Simple Parameter-efficient Fine-Tuning." CVPR 2023.
>
> >Q3: In all cases where different data budgets are compared (Tables 1,5,7, and 10), the performance is worse for larger budgets consistently across methods.
>
> A3: Data budget in D$^2$C and our paper refers to the size of the subset that is selected for efficient training. The observed trend is a result of fixed-iteration convergence lag. With a constant batch size and iteration count (100K), a larger data budget (e.g., 100K vs 10K images) drastically reduces the number of "epochs" or updates per sample. Specifically, for a 10K subset, each sample is seen ~1280 times, whereas for a 100K subset, it is seen only ~128 times. Although larger budgets provide a higher performance ceiling, they require significantly more iterations to reach it. In the early stages of training, with the same number of iterations, smaller data budgets can perform better if the selection is well-optimized, as each sample is updated more frequently. This confirms our method’s core advantage: accelerating convergence by focusing the training process on a geometry-consistent core manifold.

---

> > ### Author Rebuttal · Reviewer_FdRz · 2026-04-03
> >
> > I appreciate the detailed response by the authors.
> > The answer to Q2 resolves my issue, as it makes sense to me that comparing to performance on the full dataset is prohibitively expensive, and the results in paper compare convincingly to other condensation methods.
> >
> > While the answers to Q1 and Q3 also make sense on their own, there appears to be somewhat of an implicit contradiction which raises further question.
> > The (up to) 100x acceleration mentioned in the answer to Q1, seems to be based on the assumption that each images is seen by the network the same number of time, causing the training time to be roughly proportional to the size of the dataset.
> > In the answer to Q3, however, the total number of iterations is kept fixed, therefore the training time should remain essentially the same. Of course, in this case a noteworthy improvement in performance is observed, making it useful for a different reason.
> >
> > Both, while originally reading the paper, as well as while reading the rebuttal, it was unclear to me what the intended purpose of the data condensation method was. Both seem sensible goals to me, and as such this seems more of an issue with clarity of presentation. Thus some further clarification is needed.
> >
> > Nonetheless, I believe the current response justifies raising the score to 4.

---

> > > ### Author Response · Authors · 2026-04-04
> > >
> > > Thank you for your feedback. The primary goal of our approach is to enable rapid training on a condensed dataset, allowing the diffusion model to quickly achieve strong performance (e.g., competitive FID scores) in a resource-efficient manner. We apologize for any confusion caused regarding the purpose of the data condensation method for diffusion model training.
> > >
> > > We would like to clarify the distinction between the answers to Q1 and Q3. In Q1, the 100$\times$ acceleration refers to the training time required to achieve the same performance (FID). As shown in Figure 1 of [1], training on the condensed set requires only 1/100th of the iterations compared to the full dataset to reach FID-50K=5.9, which leads to faster convergence. This reduction in the number of training iterations is made possible by dataset condensation and results in a substantial speed-up in training time.
> > >
> > > In Q3, we fixed the total number of training iterations at 100K to ensure a fair comparison between different condensation methods under the same training conditions. The purpose here was not to compare different data budgets, but to evaluate which condensation method performs best in terms of FID and other metrics within the same number of iterations. By fixing the iteration count, we ensured consistency across methods, making the comparison between different condensation methods clear and direct.
> > >
> > > Additionally, we will revise the problem statement to clarify: "A diffusion model trained on $\mathcal{S}$ should achieve competitive generative performance with limited training iterations."
> > >
> > > We hope this revision resolves the confusion regarding the comparison between training times and the intended purpose of our approach.
> > >
> > > [1] Huang, Rui, et al. "Accelerating Diffusion Model Training under Minimal Budgets: A Condensation-Based Perspective." CVPR 2026.

---

### Official Review · Reviewer_XqC7 · 2026-03-13

**Soundness:** 3
**Presentation:** 3
**Significance:** 2
**Originality:** 2
**Overall Recommendation:** 4
**Confidence:** 4

**Summary:**

The paper proposes a geometry-aware dataset condensation approach for training diffusion models through subset selection. The method formulates the selection problem as a distribution alignment task in feature space, using one-sided partial optimal transport (POT) to prioritize high-density regions of the full dataset while allowing unmatched mass in lower-density areas. The objective is further augmented with mean–variance feature statistics and a confidence regularizer, and optimized via a two-stage discrete procedure consisting of greedy subset construction followed by swap-based refinement. Experiments on ImageNet with DiT and SiT backbones show substantial improvements over prior subset selection baselines, including D²C, across different data budgets and resolutions, along with favorable runtime and supporting ablation studies.

**Compliance With Llm Reviewing Policy:**

Affirmed.

**Final Justification:**

The authors have provided additional clarifications and experimental evidence in their rebuttal, which have sufficiently addressed my main concerns. Based on the improved clarity and supporting evidence, I am willing to raise my score to Weak Accept.

**Key Questions For Authors:**

* Some reported gains are very large, for example, at 512×512 with only 10K images (10/class), the FID is surprisingly low. This amplifies the need for transparent and uniform training/evaluation settings across all methods, ideally re-running baselines under the exact same pipeline.
* Ablations suggest each component contributes, though margins between balanced OT and one-sided POT at 10K are modest (e.g., FID 3.54 vs 3.43). Therefore, It would be helpful to further discuss the benefits of one-sided POT over balanced OT and provide insights into why the observed differences are relatively small.
* To better isolate the contribution of this design choice, it would be helpful if the authors could provide an ablation study comparing different OT variants within the same framework.
* A closely related line of work considers pruning followed by training-time reweighting to improve data-efficient diffusion training. For example, Pruning then Reweighting [1] first selects a subset using feature-based importance scores and then applies class-wise distributionally robust optimization (DRO) to reweight samples during diffusion training. In contrast, the proposed method focuses on geometry-aware subset selection via partial optimal transport but does not incorporate reweighting during training. A direct comparison would help clarify the relative benefits of improving subset quality versus correcting distribution shift through reweighting.

[1] (ICASSP 25') Pruning then Reweighting: Towards Data-Efficient Training of Diffusion Models

**Strengths And Weaknesses:**

**Strengths**
* **Clear motivation**: The high-level motivation for geometry-aware selection is well-written, with clear intuition for why balanced OT is suboptimal in the extreme budget regime.

* **Sufficient experimentl results**: Broad evaluation across architectures (DiT-L/2, SiT-L/2), budgets (10K/50K/100K), and resolutions (256 and 512), with multiple generative model evaluation metrics (FID, IS, Precision, Recall). Ablations on each component (one-sided POT vs balanced OT, removal of statistics and confidence terms) and sensitivity to κ and dummy-source parameters suggest robustness help to understand the effect of the proposed comonents.

**Weakness**
* **Weak Novelty**: The framework primarily combines several existing components, including OT-based distribution alignment, partial OT relaxation, and greedy-style subset optimization. While the integration of these elements is reasonable, it is not entirely clear whether the proposed formulation introduces fundamentally new algorithmic insights beyond adapting established techniques to the dataset condensation setting. In particular, OT-based distribution matching has been widely used in data condensation and distillation, and partial OT formulations have already been explored in related contexts to handle capacity mismatches. Similarly, greedy construction followed by local refinement is a common strategy in subset selection problems.

* **Missing related work or comparisons:** Limited discussion and empirical comparison to unbalanced OT (UOT) formulations and alternative relaxations (e.g., Sinkhorn divergence), which are closely related to POT in spirit. Recent work on pruning + reweighting pipelines for diffusion training (class-wise DRO after selection) is not discussed or compared.

---

> ### Author Rebuttal · Authors · 2026-03-30
>
> >Q1: About novelty.
>
> A1: Our work introduces new insights in the following areas:
>
> - **New Frontier in Generative Task**: Dataset condensation for diffusion training remains highly unexplored. We identify that the core challenge of this task is the extreme sensitivity of the diffusion ELBO to data support distortions.
>
> - **Unified Discrete Alignment Framework**: We move beyond heuristic ranking strategies by proposing a principled, geometry-aware distribution matching methodology, optimizing local geometry, global statistics, and semantic consistency to directly align with the diffusion likelihood objective.
>
> - **Diffusion-Free Selection**: A critical limitation of existing approaches is their reliance on high-quality pre-trained diffusion models to guide data selection. This is often impractical since a common goal of condensation is to facilitate training such models in new domains. Our selection method remains entirely diffusion-free.
>
> - **Discovery of Geometric Anchor Distortion**: We reveal that balanced alignment introduces bias by including peripheral outliers. One-sided alignment acts as a "geometric filter," ensuring that limited subset capacity focuses on the core manifold, thus preventing generative bias.
>
> >Q2: About training/evaluation settings.
>
> A2: All methods are evaluated under a unified training and evaluation pipeline. The D$^2$C results were directly cited from the original paper, while all other baselines, including our method, were trained and tested using the D$^2$C source code, following the evaluation protocol from guided-diffusion [1]. The only difference is the selection procedure, which highlights our advantage. To further verify fairness, we reran D$^2$C for Tables 1 and 2 and obtained results similar to those reported in the original paper.
>
> The modest difference in FID at 512×512 with 10K images is due to our method's faster convergence. In the D$^2$C paper, FID scores after 300K iterations for DiT-L/2 and SiT-L/2 are 5.8 and 4.22, respectively, but at 100K iterations, they had not fully converged, explaining the larger gap at this point. This highlights our method's faster convergence.
>
> [1] Dhariwal, Prafulla, and Alexander Nichol. "Diffusion Models Beat GANs on Image Synthesis." NeurIPS 2021.
>
> >Q3: About the benefits of one-sided POT over balanced OT.
>
> A3: The advantage of one-sided POT lies in its ability to suppress unreliable transport to low-density regions and focus the subset on geometrically stable support. This is quantitatively shown in Tables 22/23, where POT significantly reduces the mass in noisy bins compared to balanced OT. Under the 10K budget, the attach phase and our strong regularization partially compensate for imperfections in the initial subset, reducing the visible difference between balanced OT and one-sided POT. Even in this compensated setting, POT still shows a clear gain in Precision (0.78 vs. 0.75), indicating better alignment with the clean data manifold. This advantage becomes more pronounced as the attach phase is weakened or removed, with performance relying more on the intrinsic quality of the selected subset, as shown below:
> |Attach Encoder|Balanced OT|POT|
> |-|-|-|
> |None|8.92|7.36|
> |MAE-L|7.29|6.21|
>
> >Q4: About the comparison of different OT variants.
>
> A4:  We conducted an ablation study comparing different OT variants within the same subset-selection pipeline, with all other components remaining unchanged. The results show that improving performance over balanced OT is not straightforward: Unbalaced OT may cause distribution drift due to excessive relaxation, and the one-sided KL-relaxed OT uses blurred transport allocation. In contrast, our one-sided POT achieves better results by performing hard partial matching, preserving the data distribution’s geometric structure and delivering superior performance.
> |Method|FID|IS|Prec.|Rec.|
> |-|-|-|-|-|
> |UOT|5.31|403.0|0.63|0.30|
> |1-sided KL-OT|3.52|413.0|0.75|0.28|
> |Ours|3.43|414.3|0.78|0.28|
>
> *Note that Sinkhorn divergence is equivalent to balanced OT in our formulation.*
>
> >Q5: About the comparison with "Pruning then Reweighting".
>
> A5:  We evaluated "Pruning then Reweighting" (PTR) by replacing our POT selection with Gaussian pruning and applying its reweighting logic, ensuring all other components, including the attach phase, remained identical.
> |Method|Eval. Samples|FID|IS|Prec.|Rec.|
> |-|-|-|-|-|-|
> |PTR|10K|12.79|381.9|0.93|0.31|
> |Ours|10K|5.76|430.2|0.78|0.69|
> |PTR|50K|10.05|382.8|0.93|0.08|
> |Ours|50K|3.43|414.3|0.78|0.28|
>
> Reweighting acts only as a post-hoc fix for distributional shift and cannot recover critical manifold support or modes omitted during selection. This is reflected in the severe mode collapse of PTR, which yields a Recall of only 0.08. Our method avoids reweighting because each sample in the condensed set is given equal weight during the matching process. Reweighting would disrupt the uniform distribution and break the geometry alignment achieved through OT-based matching.

---

> > ### Author Rebuttal · Reviewer_XqC7 · 2026-04-03
> >
> > I would like to thank the authors for their detailed response and the inclusion of additional experimental results. The rebuttal addresses some of my concerns by clarifying the intended role of one-sided alignment and by providing additional comparisons. However, I remain unconvinced that the paper establishes a sufficiently strong algorithmic novelty beyond a careful integration of existing ingredients. In particular, the response still emphasizes the new application setting and empirical advantages, rather than clearly isolating a fundamentally new methodological insight. I also find the distinction between one-sided KL-relaxed OT and the proposed one-sided POT somewhat under-justified. Given that the reported performance gap is fairly small, the claim that KL-relaxed OT leads to “blurred transport allocation” while the proposed method better preserves geometric structure does not yet seem fully supported. Overall, some concerns have been alleviated, but the central novelty question remains.

---

> > > ### Author Response · Authors · 2026-04-04
> > >
> > > We would like to thank the reviewer for the feedback. However, there may be some misunderstandings that we would like to clarify.
> > >
> > > >About algorithmic novelty.
> > >
> > > We respectfully disagree with the assessment regarding novelty. Our work moves beyond heuristic ranking strategies by identifying and resolving the geometric anchor distortion problem through a principled, geometry-aware distribution matching methodology. For diffusion training, we found that two requirements naturally arise from its ELBO objective: (1) geometric alignment, as ELBO-based diffusion training is sensitive to structural bias in the learned data support, and (2) distributional fidelity, since likelihood maximization requires preserving the global diversity and semantic balance of the data distribution. Based on this, we propose an alignment-driven discrete optimization framework to jointly satisfy these requirements.
> > >
> > > Regarding specific techniques, previous OT-based dataset distillation works focus on classification tasks, where the goal is to match decision boundaries. However, in generative modeling, decision boundaries are not suitable, especially in high-dimensional and small data budgets. This is not just a "setting change" but a structural flaw that biases the generative ELBO. Our approach, using one-sided partial OT with a dummy-source reformulation, specifically addresses this by filtering out low-density regions and ensuring alignment with the core manifold, which is crucial for stable generative model training. To the best of our knowledge, no prior work has instantiated a dummy-source reformulation of one-sided partial OT as presented in our paper, and we provide a detailed derivation in **Appendix B**, ensuring efficiency in subset selection. Additionally, our batched parallel swap-based refinement searches the full 1-swap space by comparing each selected point against all unselected candidates, rather than relying on restricted neighborhood exchanges. This yields a solution with full 1-swap stability over the entire ground set, making the optimization more robust than standard greedy or local-swap strategies. We further provide a rigorous upper-bound analysis in **Appendix C** to support the theoretical soundness of this refinement procedure.
> > >
> > > >About comparison with one-sided KL-relaxed OT.
> > >
> > > From the previous table, we can see that one-sided KL-relaxed OT does not achieve better results than balanced OT. The gap between one-sided KL-relaxed OT and POT becomes substantially wider when the Attach Phase is removed or weakened. As shown in the Table below, our one-sided POT consistently outperforms one-sided KL-relaxed OT by a significant margin, across different settings. This performance improvement represents a major leap in the quality of the generative training substrate and demonstrates the robustness of POT in challenging conditions.
> > >
> > > |Attach Encoder|1-sided KL-OT|Our POT|
> > > |:-:|:-:|:-:|
> > > |None|8.79|7.36|
> > > |MAE-L|7.58|6.21|
> > >
> > > This substantial improvement can be further supported by examining the distribution of mass across different density bins (we use the same setting as Table 22 (Appendix) to report the comparison). One-sided KL-relaxed OT tends to "collapse" mass into specific modes. For example, it allocates 0.1227 (12.27%) of the total mass to the bin [0.0869, 0.0979], causing sharp jumps between adjacent bins. The relative masses across consecutive bins are: 0.0001, 0.0016, 0.1227, 0.0098, and 0.0612, highlighting these abrupt changes. This may lead to instability in training and a loss of geometric diversity.
> > > |Density Range|Number of Real Samples|POT|1-sided KL-OT|
> > > |:-:|:-:|:-:|:-:|
> > > [0.0654,0.0763]|65|0.0424|0.0001|
> > > |[0.0763,0.0869]|65|0.0517|0.0016|
> > > |[0.0869,0.0979]|65|0.0527|0.1227|
> > > |[0.0979,0.1066]|65|0.0527|0.0098|
> > > |[0.1066,0.1139]|65|0.0527|0.0612|
> > >
> > > In contrast, POT provides a more stable redistribution of mass by retaining the stable characteristics of balanced OT while reducing the matching in low-density, unstable regions. As a result, POT is more effective at maintaining the integrity of the data manifold’s structure compared to one-sided KL-relaxed OT.
> > >
> > > The mechanism behind this difference can be attributed to the way each method handles target-side relaxation. One-sided KL-relaxed OT in this implementation is unstable because the target-side relaxation is realized through global multiplicative reweighting, not explicit partial-capacity control. This method makes neighboring bins compete for mass through the coupled Sinkhorn updates, amplifying small mismatches and resulting in irregular inter-bin oscillations. In contrast, POT avoids these issues by using explicit target capacities and a dummy reservoir for unmatched mass, yielding a much more controlled redistribution. This allows POT to maintain a stable and meaningful mass distribution, particularly in the presence of sparse regions, ensuring the model's robustness and stability during training.

---

### Decision · Program_Chairs · 2026-04-30

**Decision:**

Accept (regular)

**Comment:**

This paper proposes a geometry-aware dataset subset selection approach for diffusion model training using one-sided partial optimal transport. After rebuttal, all reviewers gave positive scores. The authors' rebuttal effectively addressed key concerns through ablations demonstrating the method's independence, comprehensive hyperparameter sweeps confirming robustness, and cost analysis justifying the impracticality of full-dataset comparison. Reviewers acknowledged the experiments across different architectures (DiT, SiT), resolutions, and budgets, and the coherent formulation of subset selection as distribution alignment.

As the authors acknowledged during discussion, the camera-ready should incorporate the promised revisions: clarifying the relationship between the acceleration claim and fixed-iteration evaluation protocol, refining the usage of "geometry" to precisely denote distributional support geometry in representation space, and explicitly discussing the method's scope within small-budget regimes.